# Valid Selection among Conformal Sets

**Mahmoud Hegazy**[1,2]    **Liviu Aolaritei**[3]    **Michael I. Jordan**[2,3]    **Aymeric Dieuleveut**[1]

[1]CMAP, École polytechnique, Institut Polytechnique de Paris
[2]Inria, École Normale Supérieure, PSL Research University
[3]Department of Electrical Engineering and Computer Sciences, University of California, Berkeley

mahmoud.hegazy@polytechnique.edu   liviu.aolaritei@berkeley.edu
jordan@cs.berkeley.edu   aymeric.dieuleveut@polytechnique.edu

## Abstract

Conformal prediction offers a distribution-free framework for constructing prediction sets with coverage guarantees. In practice, multiple valid conformal prediction sets may be available, arising from different models or methodologies. However, selecting the most desirable set, such as the smallest, can invalidate the coverage guarantees. To address this challenge, we propose a stability-based approach that ensures coverage for the selected prediction set. We extend our results to the online conformal setting, propose several refinements in settings where additional structure is available, and demonstrate its effectiveness through experiments.

## 1   Introduction

Conformal Prediction (CP) provides a principled framework for uncertainty quantification by constructing prediction sets with guaranteed marginal coverage [1–3]. For a desired coverage level $1 - \alpha$, a conformal predictor outputs a set that on average contains the true label with at least this probability. The appeal of conformal prediction lies in its minimal assumptions about the data distribution and the underlying predictive model. In practice, multiple conformal prediction algorithms may be available for a given task, arising from variations in underlying models or data splits. This multiplicity motivates the choice of the most desirable set, often the smallest. To illustrate, consider a prediction problem with feature $X$ and $K$ sets $\{C_i^\alpha(X)\}_{i=1}^K$, each generated by a different conformal predictor. Suppose each $C_i^\alpha(X)$ satisfies the marginal coverage guarantee $\mathbb{P}\{Y \in C_i^\alpha(X)\} \geq 1 - \alpha$, for all $i \in [K]$, where $Y$ denotes the label associated to $X$. Although each $C_i^\alpha(X)$ individually meets the coverage guarantee, selecting the smallest set generally invalidates the guarantee due to dependencies on the data introduced by the selection.

**Contributions and outline.** To address this issue, we first introduce a novel perspective on the selection process based on algorithmic stability [4, 5]. The core idea is to employ a *stable* randomized selection mechanism, meaning its output is robust to small input perturbations. Such stability then allows us to transfer the marginal coverage of *individual conformal predictors* to the selected set. We introduce several stable selection rules, in particular MinSE, which we prove to be optimal. We further extend approach, by introducing an adaptive and a derandomized variant. These contributions are given in Section 3, after recalling preliminaries on conformal prediction in Section 2.

Furthermore, we extend in Section 4 our work to the online conformal setting [6–9], where data arrives sequentially, and predictions are made in real-time. We first demonstrate how our stability-based approach integrates seamlessly with existing online conformal methods, particularly with the approach of [10], to enable more adaptable selection among online predictors. Finally, in Section 5, we explore methods to optimize the implementation of our stable selection framework in practice. In particular, we take a closer look at the split conformal setting, where the stability-based bound can be overly conservative, and propose a recalibration mechanism that can achieve better empirical performance. Lastly, we validate our approaches on multiple experimental settings in Section 6.

39th Conference on Neural Information Processing Systems (NeurIPS 2025).

**Related Work.** The motivation to select the smallest conformal prediction set has spurred a series of recent works introducing principled selection methods that retain coverage guarantees while favoring smaller sets. For example, Liang et al. [11] proposes to merge multiple predictors by selecting the one with smallest average set size on the calibration data. Moreover, [12] proposes split-conformal methods that either inflate quantiles or use independent splits to preserve coverage after selection. Building on this approach, for the classification setting, Luo and Zhou [13] proposes a method for constructing the best possible conformal score, which may be expressed as a weighted average of different scores. In contrast to such methods, our work enables *pointwise* selection depending on each $X$ while maintaining validity, i.e. our aim is not to select the predictor with best *average performance*, but rather to pointwise select a predictor producing a small set for *each* realization of $X$.

In the online setting, Gasparin and Ramdas [10] proposed a method for conformal online model aggregation, which adapts model weights over time based on past performance. Their method combines prediction sets using a weighted majority vote with the weights learned in an online fashion. In another recent contribution, Hajihashemi and Shen [14] handles distribution shifts but focuses on temporal adaptation to distribution shifts, our approach addresses the distinct challenge of ensuring valid coverage when selecting across multiple predictors in static or online settings.

Our approach builds on algorithmic stability, a concept with roots in generalization properties of algorithms [15, 16] and differential privacy [17, 18]. Originally introduced to ensure privacy-preserving data analysis, differential privacy has been adapted for other tasks, such as adaptive data analysis [19], where it addresses the challenges of reusing data for multiple adaptive queries. Zrnic and Jordan [4] applied stability-based techniques to establish statistical validity after selection processes. Building on their work, we extend these ideas to the conformal prediction setting, developing stability-based methods for both batch and online conformal frameworks. Within the conformal predictions literature, different notions of algorithmic stability have been leveraged. For example, Barber et al. [20] proved coverage properties of the Jackknife method under a notion of stability albeit very different from the one we use.

While the aforementioned works focus on selecting conformal predictors for a single prediction task, other research has explored related but distinct problems in the context of conformal inference. Conformalized selection methods aim to identify a subset of data points whose unobserved labels exceed a given threshold while controlling the False Discovery Rate (FDR) [21]. In a recent work, Bai and Jin [22] introduced a framework that allows data reuse for both training and selection while maintaining finite-sample FDR control. Although it addresses a different problem than ours, the focus on managing data reuse aligns conceptually with our goal of ensuring valid coverage despite dependencies introduced by the selection.

## 2  Preliminaries in Conformal Prediction

We consider CP in two key scenarios: the batch setting, which assumes i.i.d. or exchangeable samples, and the online setting, where data arrives sequentially under minimal distributional assumptions.

**Batch Setting**. We consider a dataset $\mathcal{D} = \{(X_1, Y_1), \ldots, (X_n, Y_n)\} \in (\mathcal{X} \times \mathcal{Y})^n$, where the points in $\mathcal{D}$, along with any test sample $(X, Y) \in \mathcal{X} \times \mathcal{Y}$, are assumed to be either i.i.d. or exchangeable. In the i.i.d. case, we denote by $\mathcal{P}$ the distribution from which they are drawn, and by $\mathcal{P}_X$ and $\mathcal{P}_Y$ its marginals over $X$ and $Y$, respectively. Without any further assumptions on the data generating process, conformal prediction allows to construct a (random) prediction set $C^\alpha(X)$ with the guarantee

$$\mathbb{P}\{Y \in C^\alpha(X)\} \geq 1 - \alpha. \tag{1}$$

Here, the probability is taken over $(X, Y)$ as well as the randomness employed in the construction of $C^\alpha$. Arguably, the most common approach for batch conformal prediction is the *split conformal* procedure, which first partitions the dataset $\mathcal{D}$ into two disjoint subsets: $\mathcal{D}_{\text{train}} = \{(X_i, Y_i)\}_{i=1}^{n-m}$, used to train the underlying predictor $f$, and $\mathcal{D}_{\text{cal}} = \{(X_i, Y_i)\}_{i=1}^{m}$, reserved for calibration. Then, for any nonconformity score function $s$, which quantifies the error between the predictor $f$ and the true output, it estimates the empirical $\lceil (1 - \alpha)(m + 1) \rceil / m$-quantile, denoted $\hat{q}_\alpha$, of the set $\{s_i := s(X_i, Y_i, f)\}_{i=1}^{m}$. Finally, for the test point $X$, the split conformal prediction set is defined as $C^\alpha(X) := \{y \in \mathcal{Y} : s(X, y, f) \leq \hat{q}_\alpha\}$, which satisfies (1) through a rank statistic argument.

**Online Setting**. In this setting, observations $(X_t, Y_t)$ arrive sequentially for $t = 1, 2, \ldots$. At each time step $t$, we observe $X_t$ and aim to cover $Y_t$ using a prediction set $C^{(t)}(X_t)$, which is constructed

based on a base model trained on all past data $\{(X_1, Y_1), \ldots, (X_{t-1}, Y_{t-1})\}$. After making the prediction, the true label $Y_t$ is revealed, and the process continues to the next time step. Unlike the classical conformal prediction setting, where data is assumed to be exchangeable, the online setting allows for the data to be non-stationary or even adversarial. As a result, classical coverage guarantees no longer hold, and alternative notions of asymptotic coverage are required [6]. Specifically, we say that $\{C^{(t)}\}_{t \in \mathbb{N}}$ achieves asymptotic coverage if

$$\liminf_{T \to \infty} \frac{1}{T} \sum_{t=1}^{T} \mathbb{1} \left\{ Y_t \in C^{(t)}(X_t) \right\} \geq 1 - \alpha. \tag{2}$$

The limit (2) ensures that, in the long run, the fraction of instances where the true label $Y_t$ falls within the prediction set $C^{(t)}(X_t)$ meets or exceeds the desired coverage level, even under non-stationary or adversarial data. Stronger notions of asymptotic coverage can be considered, for example [23].

## 3  Smallest Confidence Set Selection

This section considers the batch setting without imposing additional restrictions, such as those in split conformal methods. Specifically, we focus on the problem of selecting the smallest among a collection of $K$ conformal prediction sets $\{C_i^\alpha(X)\}_{i=1}^{K}$. While such a selection is appealing, it inevitably invalidates the marginal coverage guarantee (1) since the selection process depends on the data. To address this issue, we develop a strategy based on algorithmic stability, ensuring adjusted coverage guarantees even after the data-dependent selection process.

### 3.1  Valid Selection via Algorithmic Stability

We first recall the notion of algorithmic stability from [4] and extend their framework to tackle the data-dependent selection in conformal prediction. We start by introducing indistinguishability.

**Definition 1** (Indistinguishability). *A random variable (r.v.) $S$ is $(\eta, \tau)$-indistinguishable from a r.v. $S_0$, denoted $S \approx_{\eta,\tau} S_0$, if for all measurable sets $\mathcal{O}$, it holds that $\mathbb{P}\{S \in \mathcal{O}\} \leq e^\eta \mathbb{P}\{S_0 \in \mathcal{O}\} + \tau$.*

This definition extends to the conditional case, denoted by $S \approx_{\eta,\tau}^{|\xi} S_0$, if the inequality holds almost surely with respect to the conditioning variable $\xi$, that is, $\mathbb{P}\{S \in \mathcal{O} \mid \xi\} \leq e^\eta \mathbb{P}\{S_0 \in \mathcal{O} \mid \xi\} + \tau$. In essence, the parameter $\eta$ measures the degree of similarity between the distributions of $S$ and $S_0$, with smaller values of $\eta$ allowing for greater similarity. Leveraging indistinguishability, we can define a notion of stability for randomized algorithms. For more precision, we define a *randomized algorithm* as a deterministic mapping from $\Xi \times \mathcal{E}$ into $\mathcal{S}$, where $\Xi$ is typically the data space, and $\mathcal{E}$ describes the inner randomness of the algorithm. We also note that the randomness of an algorithm $S$ may be either implicitly or explicitly parameterized by $\mathcal{E}$. Nonetheless, we keep the dependence on $\mathcal{E}$ explicit in order to more precisely separate different sources of randomness in our statements.

**Definition 2** (Stability). *A randomized algorithm $\hat{S} : \Xi \times \mathcal{E} \to \mathcal{S}$ is $(\eta, \tau, \nu)$-stable w.r.t. a measure $\mathcal{P}$ on $\Xi$ if there exists a r.v. $S_0$, possibly dependent on $\mathcal{P}$, such that $\mathbb{P}\{\hat{S}(\xi, \varepsilon) \approx_{\eta,\tau}^{|\xi} S_0\} \geq 1 - \nu$.*

In words, a randomized algorithm $\hat{S}$ is stable if there exists a reference r.v. $S_0$ such that, for almost any inputs $\xi \in \Xi$ (up to a probability $\nu$) sampled from the distribution $\mathcal{P}$, the distribution of $\hat{S}(\xi, \varepsilon)$ resembles that of $S_0$. Essentially, this means that, for most inputs $\xi$, the algorithm's output (that randomly depends on $\varepsilon$) behaves as if governed by a fixed distribution, independent of the specific input. We now examine how stability can be leveraged for selection among confidence sets.

Let $\zeta \in \mathcal{Z}$ and $\xi \in \Xi$ be two random variables with arbitrary dependence. Suppose that there exists a set of (possibly random) confidence intervals $\{\mathrm{CI}_s^\alpha | s \in \mathcal{S}\}$, each correlated with $\xi$ (for instance, $\xi$ may be a vector of size $|\mathcal{S}|$ containing the size of all sets $\mathrm{CI}_s^\alpha$), such that, for all $s \in \mathcal{S}$, we have

$$\mathbb{P}\left\{\zeta \notin \mathrm{CI}_s^\alpha\right\} \leq \alpha, \tag{3}$$

for some $\alpha \in (0, 1)$. Moreover, let $\hat{S}(\xi, \varepsilon)$ define an arbitrary selection algorithm. For example, $\hat{S}(\xi, \varepsilon)$ might be biased towards selecting smaller size confidence sets. Without further assumptions, individual guarantees (3) do not translate to the selected interval $\mathrm{CI}_{\hat{S}(\xi,\varepsilon)}^\alpha$, i.e. $\mathbb{P}\{\zeta \notin \mathrm{CI}_{\hat{S}(\xi,\varepsilon)}^\alpha\} \leq \alpha$ is *not* guaranteed to hold. Nonetheless, in the following theorem, we show that if $\hat{S}$ is $(\eta, \tau, \nu)$-stability, then an adjustment of the confidence level in (3) is sufficient to account for the effects of selection.

**Theorem 1** (Valid stable selection). *Assume that $\mathbb{P}\{\zeta \notin \mathrm{CI}_s^\alpha\} \leq \alpha$ holds for all $s \in \mathcal{S}$. If $\hat{S} : \Xi \times \mathcal{E} \to \mathcal{S}$ is an $(\eta, \tau, \nu)$-stable selection algorithm, then,*

$$\mathbb{P}\Big\{\zeta \notin \mathrm{CI}_{\hat{S}(\xi,\varepsilon)}^\alpha\Big\} \leq \alpha e^\eta + \tau + \nu. \tag{4}$$

## 3.2 Application to Conformal Prediction

In the context of conformal prediction, $\mathcal{S} = \{1, \ldots, K\}$ and $\{\mathrm{CI}_s^\alpha\}_{s \in \mathcal{S}}$ are the conformal prediction sets at $X$, i.e. $\{C_i^\alpha(X)\}_{i=1}^K$, where each set $C_i^\alpha(X)$ is assumed to satisfy the coverage guarantee (1). The notion of $(\eta, \tau, \nu)$-stability enables one to tackle the challenge of favoring the smallest among the $K$ conformal prediction sets $\{C_i^\alpha(X)\}_{i=1}^K$. The r.v. $\zeta$ represents the output $Y$ and we define

$$\xi := [\lambda(C_1^\alpha(X)), \ldots, \lambda(C_K^\alpha(X))], \tag{5}$$

where $\lambda(C_i^\alpha(X))$ represents a "size" (for example, a scaled Lebesgue's measure, counting measure, or more generically, any notion of set desirability) of set $C_i^\alpha(X)$, for all $i \in [K]$. Now note that $\nu$ introduced in Definition 2 is a function of the distribution of $\xi$. In line with conformal prediction methods, which benefit from distribution-free guarantees, we will focus on selection algorithms for which $\nu = 0$, and will actually obtain results that hold almost surely on $\xi$. With a slight abuse of notation, we will call these algorithms $(\eta, \tau)$-stable (or simply $\eta$-stable if, additionally, $\tau = 0$). We are ready to specialize Theorem 1 to conformal prediction.

**Corollary 1** (Smallest conformal set selection). *Let $\hat{S}$ be an $(\eta, \tau)$-stable selection algorithm (e.g., for approximating $\arg\min_{i \in [K]} \lambda(C_i^\alpha(X))$). Then, we have*

$$\mathbb{P}\Big\{Y \in C_{\hat{S}(\xi,\varepsilon)}^\alpha(X)\Big\} \geq 1 - \alpha e^\eta - \tau. \tag{6}$$

Note that the standard $1 - \alpha$ coverage can be achieved by simply adjusting the confidence level of the $K$ individual sets to $1 - (\alpha - \tau)e^{-\eta}$. In what follows, inspired by the differential privacy literature [18, 24], we provide several examples of easily implementable stable selection algorithms.

**Lemma 1** (Stability via Laplace noise). *Assume $\lambda(C_i^\alpha(X)) \in [0,1]$, for all $i \in [K]$. If $\varepsilon \sim (\mathrm{Lap}(1/\eta))^{\otimes K}$, $\varepsilon \perp \xi$, then the selection algorithm $\hat{S}$ such that $\hat{S}(\xi, \varepsilon) := \arg\min_{i \in [K]} \{\lambda(C_i^\alpha(X)) + \varepsilon_i\}$ is $\eta$-stable.*

**Lemma 2** (Stability via exponential mechanism). *Assume $\lambda(C_i^\alpha(X)) \in [0,1]$, for all $i \in [K]$. Then, the selection algorithm $\hat{S}$ with*

$$\mathbb{P}\Big\{\hat{S}(\xi, \varepsilon) = i\big|\xi\Big\} = \frac{\exp(-\eta\lambda(C_i^\alpha(X)))}{\sum_{j \in [K]} \exp(-\eta\lambda(C_j^\alpha(X)))}$$

*is $2\eta$-stable. Note that we do not need to make the distribution of $\varepsilon$ or the mapping $\hat{S}$ explicit here.*

We note that the assumption that $\lambda(C_i^\alpha(X)) \in [0,1]$ for any $i \in [n]$ was used to alleviate notations and may be relaxed up to appropriate scaling of the mechanisms. More importantly, while the two selection algorithms discussed above satisfy the stability requirement, they are adapted from the literature of differential privacy, which is strictly stronger than our required notion of stability. Thus, this additional strength can lead to overly conservative behavior when only stability is required.

To address this, we propose a new Minimum Stable Expectation (MinSE) selection mechanism, designed to achieve stability as tightly as possible. MinSE relies on a *prior* $b \in \Delta^{K-1}$, that encodes prior knowledge on which interval to select, before observing the different predictive intervals at $X$.

**Lemma 3** (Minimum stable expectation). *Let $\eta, \tau \geq 0$, and a fixed $b \in \Delta^{K-1}$, and consider the following linear program*

$$p^\star(b, \xi) = \arg\min_p \quad \sum_{i=1}^K p_i \lambda(C_i^\alpha(X))$$
$$\text{s.t.} \quad p \in \Delta^{K-1}, \ s \in \mathbb{R}_+^K, \ p_i \leq e^\eta b_i + s_i, \ \sum_{i \in [K]} s_i \leq \tau \tag{MinSE}$$

*Then, the selection algorithm $\hat{S}(\xi, \varepsilon)$ with $\mathbb{P}\{\hat{S}(\xi, \varepsilon) = i|\xi\} = p_i^\star(b, \xi)$ is $(\eta, \tau)$-stable.*

All three lemmas propose randomized selection algorithms which assign the highest probability to the smallest set. However, different from the first two, which assign nonzero probability to all the sets, MinSE assigns zero probability to the largest, whenever feasible.

To develop further intuition of MinSE, consider the case where $b = (1/K)_{i \in [K]}$ represents a uniform prior. Then, if we set $e^\eta = K, \tau = 0$, MinSE reduces to $\arg\min_i \{\lambda (C_i^\alpha (X)), i = 1, \ldots, K\}$, thus selecting deterministically the smallest set. This is reasonable, as then, for $1 - \alpha$ after selection, the original confidence sets should have $1 - \alpha/K$ coverage, which corresponds to a Bonferroni correction [25]. Now consider a less extreme example with $e^\eta = 2, \tau = 0$, and $b$ still a uniform prior. By direct checking of the feasibility conditions, one can show that MinSE will never choose any of the $\lfloor K/2 \rfloor$ largest sets. In addition, it is possible to show that MinSE achieves a notion of optimality among all stable selection mechanisms.

**Proposition 1** (Optimality of MinSE). *Let $\mathcal{A} : \Xi \times \mathcal{E} \to [K]$ be an $(\eta, \tau)$-stable algorithm w.r.t. a measure $\mathcal{P}$ on $\Xi$. Then there exists a prior vector $b \in \Delta^{K-1}$ such that it holds $\mathcal{P}$-almost surely that*

$$\sum_{i=1}^K p_i^\star(b, \xi) \lambda (C_i^\alpha(X)) \leq \sum_{i=1}^K p_i^\mathcal{A}(\xi) \lambda (C_i^\alpha(X)),$$

*where $\xi = [\lambda (C_1^\alpha(X)), \ldots, \lambda (C_K^\alpha(X))]$, the vector $p^\star(b, \xi) = (p_1^\star(b, \xi), \ldots, p_K^\star(b, \xi))$ is the output of MinSE with parameters $\eta, \tau \geq 0$ and prior $b$, and $p_i^\mathcal{A}(\xi) := \mathbb{P}_\varepsilon \{\mathcal{A}(\xi, \varepsilon) = i | \xi\}$.*

In words, for any $(\eta, \tau)$-stable algorithm, there exists a prior vector $b$ such that the distribution of the output of MinSE achieves the smallest expected size.

### 3.3 MinSE examples, tightness, and extensions

**Example 1** (Worst-case Oracles). Consider $K$ oracle confidence interval methods such that for any $X \in \mathcal{X}$, an index $j \in [K]$ is chosen uniformly at random, and oracle $j$ outputs $C_j(X) = \emptyset$, while all other oracles $i \neq j$ output $C_i(X) = \mathcal{Y}$. This setup provides a scenario to analyze the tightness of the stability guarantee. For simplicity, let $\lambda(\mathcal{Y}) > 0$ and $\lambda(\emptyset) = 0$. For any datapoint $(X, Y)$, each oracle $i$ individually has marginal miscoverage $\mathbb{P}\{Y \notin C_i(X)\} = 1/K$. Let $\exp(\eta)/K + \tau \leq 1$, applying MinSE with parameters $(\eta, \tau)$ and a uniform prior $b$, we examine the post-selection miscoverage $\mathbb{P}(Y \notin C_{\hat{S}})$. Miscoverage occurs only if the empty oracle is selected. By construction, one and only one set miscovers and has minimal size. MinSE assigns the maximum possible probability $p_j^* = \exp(\eta)/K + \tau$ to the zero-size set $j$. Thus, the miscoverage probability is $\exp(\eta)/K + \tau$. This result exactly matches the upper bound from Corollary 1, demonstrating that the stability bound is indeed tight in worst-case scenarios. Arguably, this is a pathological setting due to the dependence structure of the confidence sets. Nonetheless, we empirically show in the next example that the stability coverage bound is tight, even with independent confidence sets.

**Example 2** (Coin flips). Let $\mathcal{Y} = [0, 1]$ and each of the $K$ sets be constructed using an independent coin flip: for each $i \in [K]$, $C_i^\alpha(X)$ is equal to $\mathcal{Y}$ with probability $1 - \alpha$ and to $\emptyset$ with probability $\alpha$. In this example, we appropriately adjust the coverage of the original coin flips to achieve coverage of $1 - \alpha$ after $\eta$-stable selection. Contrary to Example 1, here the different oracles are independent. As shown in Figure 1, the $\eta$-stable selection using MinSE almost exactly achieves $1 - \alpha$ coverage across different stability levels $\eta$, particularly as $K$ grows. This suggests that while the stable selection mechanism inflates the probability of yielding the full set $\mathcal{Y}$, selecting the set with the minimum size effectively counterbalances this inflation. Therefore, even with oracle independence, without additional assumptions, the inflation of the original sets is necessary.

**Example 3** (Toy regression model). Let $Y = |X| + \mathcal{N}(0, 0.25)$, with $X \sim \text{Uniform}([-1, 1])$, and consider the following two predictors $f_1(X) = X$ and $f_2(X) = -X$. This setup could arise, for example, when the data is split during training, e.g., due to privacy or design constraints, and is meant to be favourable for our method. Indeed, on each half-space ($X \geq 0$ or $X \leq 0$), one confidence interval is much smaller than the other one. Numerical results, given to the right of Figure 1, show that despite the inflation in set size, stable selection leads to an improvement in the average set size, in comparison to relying on any individual predictor, while guaranteeing the same coverage. Numerically, the individual sets are about 30% wider on average. This improvement highlights the advantage of using stable selection in settings where different predictors have complementary strengths, i.e. are accurate on different subsets of $\mathcal{X}$.

**Remark 1** (Adaptive version—AdaMinSE). A practical consideration when using MinSE is the selection of the stability parameters $\eta$ and $\tau$. Given $K$ conformal predictors, each achieving at

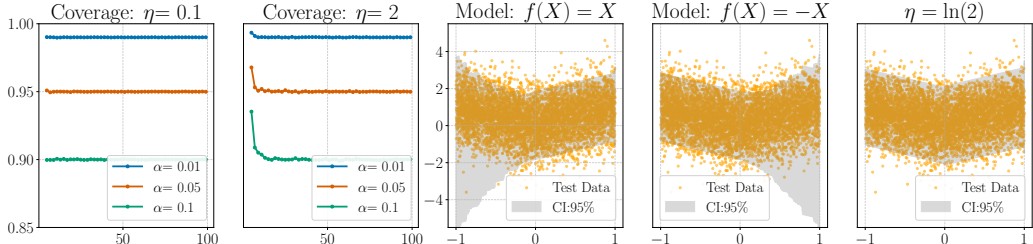

Figure 1: (Left-example 2) Coverage of the coin flips example after selection using MinSE with parameters $\eta \in \{0.1, 2\}$ and $\tau = 0$. Before selection, the oracle returns the full set $[0, 1]$ with probability $1 - \alpha \exp(-\eta)$. (Right-example 3) Second and third figures reflecting adaptive conformal intervals, with miscoverage $\alpha = 0.05$, obtained using the models $f(X) = X$ and $f(X) = -X$. Fifth figure shows the stable selection applied to the conformal sets in the first two figures, adjusted to have miscoverage $\alpha = 0.025$, using MinSE with $\eta = \ln(2)$.

least $1 - \alpha'$ coverage, and a desired post-selection coverage of $1 - \alpha$, any pair $(\eta, \tau)$ satisfying $\alpha' \leq (\alpha - \tau)e^{-\eta}$ is theoretically valid. This presents a choice, as the trade-off between $\eta$ and $\tau$ is not always immediately obvious. To address this, we introduce an adaptive version of MinSE, AdaMinSE, in Appendix A.1. This method automatically optimizes the $(\eta, \tau)$ trade-off to achieve the target $1 - \alpha$ coverage after selection, thereby alleviating the need for manual parameter tuning.

**Remark 2** (Derandomization). The randomized output of the stable selection mechanisms, such as MinSE, may be undesirable in certain applications where deterministic prediction sets are required. To address this limitation, it is possible to derandomize the selection process. Building upon techniques from [26], one can construct a single, deterministic prediction set from the output probabilities of the stable selection rule, while still preserving a (correspondingly adjusted) coverage guarantee. The precise results are deferred to Appendix A.2.

**Remark 3** (Conditional Coverage). For clarity and ease of exposition, our main analysis has focused on marginal coverage guarantees as defined in (1). However, the stability-based selection framework naturally extends to scenarios where the underlying conformal predictors satisfy stronger guarantees, such as conditional coverage. We provide further details on this extension in Appendix A.3.

## 4    Extension to Online Conformal Prediction

Next, we show how the framework based on algorithmic stability introduced above extends naturally to the online setting. Consider a collection of $K$ online conformal prediction algorithms that, at each time step $t \in \mathbb{N}$, produce $K$ prediction sets $\{C_i^{(t)}(X_t)\}_{i=1}^K$ for the label $Y_t$. As noted in Section 2, each prediction set depends on the entire history $\{(X_i, Y_i)\}_{i=1}^{t-1}$. Similarly to Section 3.2, we define $\xi_t := [\lambda(C_1^{(t)}(X_t)), \ldots, \lambda(C_K^{(t)}(X_t))]$, where $\lambda(C_i^{(t)}(X_t))$ denotes the size of the set $C_i^{(t)}(X_t)$, for each $i \in [K]$. The following corollary specializes Theorem 1 to this scenario, showing that an adjusted coverage guarantee can be obtained if a stable selection is applied at each time step $t$.

**Corollary 2** (Smallest online conformal set selection). *At each time $t \in \mathbb{N}$, let $\hat{S}(\xi_t, \varepsilon_t)$ be a $(\eta, \tau)$-stable selection algorithm (e.g., for approximating $\arg\min_{i \in [K]} \lambda(C_i^{(t)}(X_t))$). Assume each $C_i^{(t)}(\cdot)$ satisfies the guarantee (2). Moreover, let the elements of the sequence $(\varepsilon_t)_{t \in \mathbb{N}}$ be independent. Then,*

$$\liminf_{T \to \infty} \frac{1}{T} \sum_{t=1}^T \mathbb{P}\left\{Y_t \in C_{\hat{S}(\xi_t, \varepsilon_t)}^{(t)}(X_t)\right\} \geq 1 - \alpha e^\eta - \tau. \tag{7}$$

In essence, given multiple online conformal prediction algorithms, applying a stable selection mechanism provides a practical and systematic way to combine them. It is worth noting that while the coverage guarantee in (7) matches the one established for the batch setting in (6), the guarantee in (7) is achieved only in the long run. This distinction aligns with the nature of online conformal prediction, as discussed in Section 2. Finally, we emphasize that, unlike (2), which ensures coverage for the empirical average $\frac{1}{T} \sum_{t=1}^T \mathbb{1}\left\{Y_t \in C^{(t)}(X_t)\right\}$, the guarantee in (7) is in probability. However, the only source of randomness in this setting is the selection noise $\varepsilon_t$ in the algorithm $\hat{S}(\xi_t, \varepsilon_t)$, as all other quantities can be treated as adversarial or fixed through conditioning.

---

**Algorithm 1** Adaptive COMA (AdaCOMA)

---

**Input:** $K$ conformal algorithms $\{C_i^{(t)}\}_{i=1}^K$, stability parameter $(\eta, \tau)$, initial weights $w^{(1)} = (1/k, \ldots, 1/k)$

**For:** $t = 1, 2, \ldots$

Compute $w^{(t)}$ using COMA.

Compute $p^\star((w^{(t)}), \xi_t) \in \Delta^{K-1}$ using MinSE with $b = w^{(t)}$ and parameters $(\eta, \tau)$

**Output:** Any of the following two options:

Option 1: Combined set $C_{\text{comb}}^{(t)}(X_t)$ equal to $\left\{ y \in \mathcal{Y} \big| \sum_{i=1}^K p_i^\star(w^{(t)}, \xi_t) \mathbb{1}\left\{ y \in C_i^{(t)}(X_t) \right\} \geq \frac{1}{2} \right\}$

Option 2: Combined predictor leading to $C_{\hat{S}_{(\xi_t, \varepsilon_t)}}^{(t)}(X_t)$, with $\mathbb{P}\left\{ \hat{S}_{(\xi_t, \varepsilon_t)} = i \big| \xi_t \right\} = p_i^\star(w^{(t)}, \xi_t)$

---

### 4.1 Adaptive Conformal Online Model Aggregation.

Conformal Online Model Aggregation (COMA) [10] extends online conformal prediction by addressing the challenge of model aggregation. It combines prediction sets from multiple algorithms through a voting mechanism, where weights are dynamically adjusted over time based on past performance. Formally, at each time step $t$, COMA assigns weights $w^{(t)} = [w_1^{(t)}, \ldots, w_K^{(t)}] \in \Delta^{K-1}$, which reflect the relative importance of each of the $K$ underlying conformal predictors according to the following rule $w_i^{(t)} \propto \exp\left( -\gamma_{t-1} \sum_{j=1}^{t-1} \lambda\left( C_i^{(t)}(X_j) \right) \right)$, where $\gamma_t$ is the adaptive learning rate from the AdaHedge algorithm [27], an adaptive version of the Hedge algorithm [28]. COMA then outputs the aggregated prediction set $C^{(t)} := \left\{ y \in \mathcal{Y} \big| \sum_{i=1}^K w_i^{(t)} \mathbb{1}\{ y \in C_i^{(t)}(X_t) \} \geq \frac{1}{2} \right\}$, which can be interpreted as the aggregated set obtained by selecting the $i$-th conformal set with probability $w_i^{(t)}$.

**Non-adaptiveness of COMA.** Crucially, at time $t$, the COMA framework assigns weights to the $K$ conformal algorithms using only the observations up to time $t-1$, *without* access to $X_t$ or the prediction sets $\{C_i^{(t)}(X_t)\}_{i=1}^K$. Furthermore, due to its AdaHedge formulation, COMA optimizes the weights on average over time, without adapting to each individual $X_t$.

**AdaCOMA.** To achieve the best of both worlds, we incorporate COMA into our stable selection algorithm. Specifically, at iteration $t$, we use COMA's weights $w^{(t)}$ as the prior for a stable selection mechanism, which selects after observing both $X_t$ and the sets $\{C_i^{(t)}(X_t)\}_{i=1}^K$, allowing for pointwise adaptability. The combined procedure, termed AdaCOMA, is detailed in Algorithm 1.

COMA does not have assumption-free coverage guarantees for a fixed target level. However, letting

$$\beta_t := \mathbb{E}\left[ \sum_{i=1}^K w_i^{(t)} \mathbb{1}\left\{ Y_t \notin C_i^{(t)}(X_t) \right\} \right], \tag{8}$$

Gasparin and Ramdas [10] show that the following holds

$$\mathbb{P}\left\{ Y_t \notin C^{(t)} \right\} \leq 2\beta_t. \tag{9}$$

In the remainder of this section, we analyze the coverage guarantees of AdaCOMA in comparison to the bound in (9) for COMA. Detailed bounds on $\beta_t$ under additional assumptions, are given in Gasparin and Ramdas [10].

**Proposition 2** (Adaptive COMA). *Consider Algorithm 1, and let $\beta_t$ be defined as in* (8)*. Then,*

- *the set $C_{\text{comb}}^{(t)}(X_t)$ satisfies $\mathbb{P}\left\{ Y_t \in C_{\text{comb}}^{(t)}(X_t) \right\} \geq 1 - 2\left( \beta_t e^\eta + \tau \right)$,*

- *the set $C_{\hat{S}_{(\xi_t, \varepsilon_t)}}^{(t)}(X_t)$ satisfies $\mathbb{P}\left\{ Y_t \in C_{\hat{S}_{(\xi_t, \varepsilon_t)}}^{(t)}(X_t) \right\} \geq 1 - \beta_t e^\eta - \tau$.*

Proposition 2 demonstrates that AdaCOMA inherits the flexibility of COMA while improving its adaptability to current observations through the stable selection mechanism.

# 5 Post-selection Calibration in Split Conformal Prediction

The coverage bounds derived from our stability-based approach (e.g., Corollary 1) are distribution-free and hold under minimal assumptions, allowing them to extend to even the adversarial online case. While potentially tight in worst-case scenarios, these bounds can be conservative when additional structure is available, particularly in the batch setting. By leveraging the inherent rank structure of the split conformal method, this section develops a recalibration procedure specifically for split conformal prediction, aiming to achieve tight finite-sample coverage guarantees after selection.

We operate within the standard split conformal setup introduced in Section 2, with a calibration dataset $\mathcal{D}_{\mathrm{cal}} = \{(X_i, Y_i)\}_{i=1}^m$ and a test point $(X, Y)$. We assume the sequence of $m + 1$ data points $\{(X_1, Y_1), \ldots, (X_m, Y_m), (X, Y)\}$ to be exchangeable. We consider $K$ base predictors $f_1, \ldots, f_K$ and corresponding non-conformity score functions $s_1, \ldots, s_K$. For each base predictor $k \in [K]$ and datapoint $i \in [m]$, denote the non-conformity scores as $s_{k,i} \coloneqq s_k(X_i, Y_i, f_k)$. We use $s_{k,(r)}$ to denote the $r$-th order statistic of $s_{k,1}, \ldots, s_{k,m}$. Finally, for any $k$ and $i$, we denote the rank of a score $s_{k,i}$ as $R_{k,i} \coloneqq \sum_{j=1}^m \mathbb{1}\{s_{k,j} \leq s_{k,i}\}$. Using these definitions, we can parameterize the conformal prediction sets using ranks, i.e., for any rank index $R \in [m]$, we define

$$C_k(X, R) \coloneqq \left\{ y \in \mathcal{Y} : s_k(X, y, f_k) \leq s_{k,(R)} \right\},$$

which recovers the classical set $C^\alpha$ recalled in Section 2 for $R_\alpha = \lceil (1 - \alpha)(m + 1) \rceil$, satisfying, for any $k \in [K]$, $\mathbb{P}\{Y \in C_k(X, R_\alpha)\} \geq 1 - \alpha$.

**Calibration After Selection using Effective Ranks.**  We now introduce an arbitrary (stochastic) selection rule $\hat{S}$. Our goal is to determine an *effective rank* $\hat{R}_\alpha$ such that a similar property holds, but *for the selected interval*, i.e., $\mathbb{P}\{Y \in C_{\hat{S}(X,\varepsilon)}(X, \hat{R}_\alpha)\} \geq 1 - \alpha$. To that end, we use a recalibration process after selection, that uses the effective ranks *as meta-scores*. For each point $i \in [m]$, we apply the selection rule using its feature vector $X_i$ and *independent* randomness $\varepsilon_i \sim \mathcal{P}_\varepsilon$. We now define the effective rank (or the meta-score) for the $i$-th point as:

$$\hat{R}_i \coloneqq R_{\hat{S}(X_i, \varepsilon_i), i},$$

that is, the rank of the $i$-th point's score calculated using the selected predictor $\hat{S}(X_i, \varepsilon_i)$. Subsequently, we define the sequence of effective ranks $\mathcal{R} \coloneqq (\hat{R}_1, \ldots, \hat{R}_m)$. Using uniform random tiebreaks between equal ranks in $\mathcal{R}$, for $t \in [m]$, we use $\hat{R}_{(t)}$ to denote the $t$-th order statistics of $\mathcal{R}$.

**Theorem 2.** *Assume that $\hat{S}$ is independent of $\mathcal{D}_{\mathrm{cal}}$. Let $\tau_\alpha = \lceil (1 - \alpha)(m + 1) \rceil \leq m$. Then,*

$$\mathbb{P}\left\{ Y \in C_{\hat{S}(X,\varepsilon)}\left( X, \hat{R}_{(\tau_\alpha)} \right) \right\} \geq 1 - \alpha.$$

Theorem 2 provides a method to maintain coverage guarantees after selecting among multiple split conformal predictors. The use of effective ranks acts as a meta-score, allowing for the calibration of predictors even if they utilize different non-conformity score functions $s_k(\cdot)$, offering flexibility in model-specific score design. The theorem states that by selecting the appropriate order statistic $\hat{R}_{(\tau_\alpha)}$ of these effective ranks, derived using an independent selection rule $\hat{S}$, the conformal set $C_{\hat{S}(X,\varepsilon)}(X, \hat{R}_{(\tau_\alpha)})$ for the chosen predictor $\hat{S}(X, \varepsilon)$ achieves the desired coverage.

**Constructing an Independent $\hat{S}$.**  Theorem 2 mandates the selection rule $\hat{S}$ to be independent of the calibration data $\mathcal{D}_{\mathrm{cal}}$. This independence plays a critical role in the proof as it ensures that the effective ranks $\hat{R}_1, \ldots, \hat{R}_m$ are exchangeable with the unobserved effective rank of the test point. Consequently, if $\hat{S}$ aims to select the smallest set, it cannot use set sizes derived from $\mathcal{D}_{\mathrm{cal}}$-based quantiles due to the induced dependency. To ensure independence, we employ an auxiliary dataset $\mathcal{D}_{\mathrm{aux}}$, disjoint from and independent of $\mathcal{D}_{\mathrm{cal}}$. For each predictor $k \in [K]$ and test point $X$, proxy quantiles $\hat{q}_{\alpha,k}^{\mathrm{aux}}$, computed from $\mathcal{D}_{\mathrm{aux}}$ (at a preliminary miscoverage rate $\tilde{\alpha}$), define proxy conformal sets $C_k^{\mathrm{aux}}(X)$ and their corresponding sizes $\lambda(C_k^{\mathrm{aux}}(X))$. The vector of these proxy sizes, $\xi^{\mathrm{aux}}(X) \coloneqq [\lambda(C_1^{\mathrm{aux}}(X)), \ldots, \lambda(C_K^{\mathrm{aux}}(X))]$, is then independent of $\mathcal{D}_{\mathrm{cal}}$. A selection rule $\hat{S}(X, \varepsilon)$ based solely on $X$ and this $\xi^{\mathrm{aux}}(X)$ (e.g., employing stable mechanisms from Section 3.2, such as MinSE) thus satisfies the independence condition. This allows Theorem 2 to be applied directly for

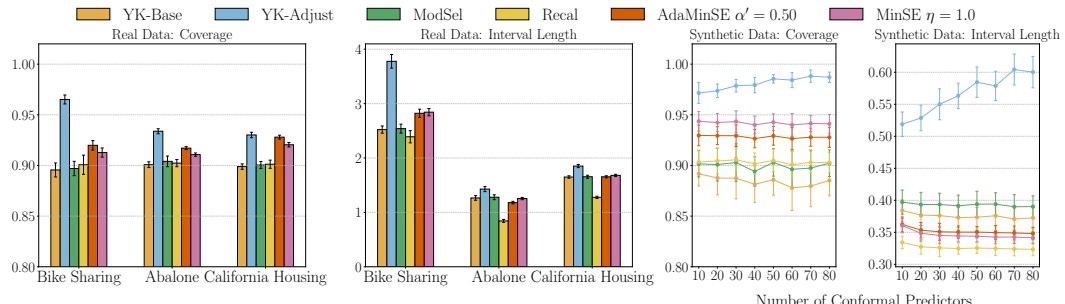

Figure 2: Marginal coverage and average lengths, for real datasets (Bike Sharing, Abalone, California Housing) and synthetic data; Error bars represent twice the standard error of mean estimation using multiple seeds for real datasets and 2 s.d. for the synthetic data.

$1 - \alpha$ coverage, avoiding the inflation factors inherent in stability-based bounds that depend on $\mathcal{D}_{\text{cal}}$. The practical effectiveness of such selection hinges on how accurately the proxy information from $\mathcal{D}_{\text{aux}}$ reflects the true characteristics based on $\mathcal{D}_{\text{cal}}$.

## 6 Experiments

To illustrate our approach in practice, we present here two simple experimental setups, one on synthetic and one on real data. We defer online experiments, additional batch experiments, and further experimental details to Appendix C. We compare our approach to Yang and Kuchibhotla [12] and Liang et al. [11], denoted as YK and ModSel, respectively. As Yang and Kuchibhotla [12] proposed multiple algorithms, adopting the following naming convention from Liang et al. [11], we compare against YK-Adjust and YK-Base. YK-Adjust adjusts the underlying conformal predictors to ensure valid coverage after selecting the best-on-average conformal predictor on the calibration dataset. YK-Base simply selects the best-on-average conformal predictor and does not have coverage guarantees. We report the performance of MinSE with parameters ($\eta = 1, \tau = 0$) and AdaMinSE with $\alpha' = 0.05$. In addition, we report the performance of Recal, which is based on AdaMinSE with $\alpha' = 0.02$, followed by the recalibration procedure of Section 5. For Recal, the calibration dataset is further split into two blocks to construct $\mathcal{D}_{\text{cal}}$ and $\mathcal{D}_{\text{aux}}$, so that the selection satisfies the independence requirement of Theorem 2. For the experiments presented here, we target a miscoverage level of $\alpha = 0.1$. Except for YK-Base, all methods guarantee miscoverage $\alpha = 0.1$ after selection.

Throughout the experiments, we aim to design a scenario where the performance of individual predictors varies across the input space. As such, using clustering, we split the feature space into 5 disjoint sets and train each predictor exclusively on a randomly selected subset. We provide additional experiments without such data splitting in Appendix C. Moreover, the code to reproduce the experiments is available in the supplementary material[1].

**Synthetic Regression**. We generate $n$ data points, $\{(X_i, Y_i)\}_{i=1}^n$, with $X_i \sim \mathcal{N}(0, I_d)$ (before feature space splitting), and the response variable defined as $Y_i = \sin(\langle \beta, X_i \rangle) + 0.1\mathcal{N}(0, 1)$, where $\beta$ is the vector $\{1/d\}_{i \in [d]}$. The feature dimension is set to $d = 10$, and the training data is split into two blocks. In the first block, we train $K$ distinct regression models, $f_1, \ldots, f_K$, using the Kernel Ridge Regression model from scikit-learn [29]. For each model, we randomly sample the kernel function (either linear or radial basis function (RBF)) and the regularization parameter (uniformly chosen between $0.1$ and $1$). For each model $i$, we use the second block of training data to train a random forest model $g_i$ that predicts the absolute residuals $|f_i(X) - Y|$, enabling us to use the nonconformity score, defined as $s_i(X, Y) = |f_i(X) - Y|/g_i(X)$. We use 400 datapoints for the calibration dataset.

**Real Datasets**. In this experiment, we aim to model a more typical data analysis scenario. We conduct experiments on three standard regression datasets: Abalone, California Housing, and Bike Sharing [30, 31]. For each dataset, leveraging `scikit-optimize` for hyperparameter tuning [32], we used the following `scikit-learn` models: AdaBoostRegressor, DecisionTreeRegressor, GradientBoost-

---

[1]Code also available at Valid-Selection-among-Conformal-Sets.

ingRegressor, ElasticNet, RandomForestRegressor, and LinearRegression. We used $80\%$ of the data for training, $10\%$ for calibration, and $10\%$ for testing. Similar to the synthetic experiments, we used the same adaptive score function, with RandomForestRegressor trained to predict the residuals. Then, up to adjusting coverage to ensure post-selection coverage of $\alpha = 0.1$, we used the same conformal predictors for all selection methods. We normalized the labels across datasets to keep the size of conformal sets comparable.

Both synthetic and real data results are reported in Figure 2. For both experiments, YK-Adjust, MinSE, and AdaMinSE overcover, while Recal and ModSel achieve similar coverage to the target marginal coverage of $0.9$. YK-Base undercovers in the synthetic setting. Differences emerge in the resulting average interval lengths. Our proposed Recal consistently performs the best on both the synthetic and real data settings. On real datasets, MinSE and AdaMinSE are competitive with baselines on Abalone and California Housing but produce larger sets on Bike Sharing. Meanwhile, for the synthetic setting, AdaMinSE and MinSE beat the benchmarks.

## 7   Conclusion

In this paper, we introduced a stability-based framework for selecting among multiple conformal predictors while preserving coverage. By casting selection as an $(\eta, \tau)$-stable randomized mechanism, we established distribution-free guarantees that transfer the validity of individual predictors to the post-selection set, enabling pointwise (feature-dependent) selection. We instantiated this principle with practical mechanisms and proposed the MinSE mechanism, which is optimal among stable selectors, along with adaptive and derandomized variants. We further extended the framework to the online setting; combined with online aggregation, this yields AdaCOMA, which uses COMA weights as a prior for stable per-time-step selection based on the current features and sets, thereby adapting over time and across each input. Finally, for split conformal prediction, we introduced a post-selection recalibration via effective ranks that mitigates the conservativeness of worst-case stability bounds. Empirically, our methods meet the target coverage and often reduce set sizes relative to existing selection approaches across synthetic and real datasets in heterogeneous settings. Nonetheless, the stability-based guarantees are worst-case and can be conservative in benign regimes; our recalibration reduces this conservativeness but requires an auxiliary dataset, which is independent from calibration, and its effectiveness depends on the quality of proxy quantiles. Moreover, randomized selection and the choice of a stability budget introduce utility/validity trade-offs, and extending the framework by deriving tighter instant-dependent coverage bounds under additional assumptions remains an open direction.

## Acknowledgement

The work of Aymeric Dieuleveut and Mahmoud Hegazy is supported by French State aid managed by the Agence Nationale de la Recherche (ANR) under France 2030 program with the reference ANR-23-PEIA-005 (REDEEM project), and ANR-23-IACL-0005, in particular Hi!Paris FLAG chair. Liviu Aolaritei acknowledges support from the Swiss National Science Foundation through the Postdoc.Mobility Fellowship (grant agreement P500PT_222215). Additionally, this project was funded by the European Union (ERC-2022-SYG-OCEAN-101071601). Views and opinions expressed are however those of the author(s) only and do not necessarily reflect those of the European Union or the European Research Council Executive Agency. Neither the European Union nor the granting authority can be held responsible for them. This publication is part of the Chair "Markets and Learning", supported by Air Liquide, BNP PARIBAS ASSET MANAGEMENT Europe, EDF, Orange and SNCF, sponsors of the Inria Foundation.

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

# Contents

## Outline of the Appendix

This appendix provides supplementary material to the main paper. The first part details deferred content from Section 3 of the main paper: Appendix A.1 introduces and proves the Adaptive Minimum Stable Expectation (AdaMinSE) mechanism; Appendix A.2 presents and proves a derandomization technique for prediction sets; and Appendix A.3 discusses and proves an extension to conditional coverage guarantees. Appendix B provides the proofs for the theoretical results presented in Section 3 (on stable selection), Section 4 (on online conformal prediction), and Section 5 (concerning post-selection calibration in split conformal prediction). Finally, Appendix C is dedicated to additional experimental results. These results include further batch experiments under varied settings, online experiments evaluating AdaCOMA, and detailed setup information for these experiments.

## A  Deferred Content Section 3

### A.1  Adaptive Minimum Stable Expectation

One difficulty in using MinSE is tuning the stability parameters $\eta$ and $\tau$. For instance, assume a user has access to $K$ conformal predictors, each with coverage at least $1 - \alpha'$, and wishes to apply MinSE to select among them such that the coverage after selection is at least $1 - \alpha$. Then, they may choose any values of $\tau$ and $\eta$ satisfying $\alpha' \leq (\alpha - \tau)e^{-\eta}$. In particular, the utility tradeoff between $\tau$ and $\eta$ is not immediately clear.

To address this, we propose AdaMinSE, an adaptive version of MinSE, which also optimizes over the choice of $\tau$ and $\eta$. Similarly to MinSE, AdaMinSE also makes use of a prior $b \in \Delta^{K-1}$ (which can be chosen depending on the past as in Algorithm 1). However, instead of requiring the parameters $(\eta, \tau)$ as input, it simply takes the current level of miscoverage $\alpha'$ and the desired miscoverage level $\alpha$ after selection. The following proposition introduces AdaMinSE, together with its coverage guarantee.

**Proposition 3** (Adaptive Minimum Stable Expectation). *Let $\alpha', \alpha \in (0,1)$ with $\alpha' \leq \alpha$, and let $b \in \Delta^{K-1}$ be fixed. Consider the following linear program*

$$
\begin{aligned}
d^{\star}(b, \xi) = \arg\min_{d} \quad & \sum_{i=1}^{K} d_i \lambda \left( C_i^{\alpha}(X) \right) \\
\text{s.t.} \quad & d \in \Delta^{K-1}, \quad s \in \mathbb{R}_+^K, \quad \tau, \eta \geq 0, \qquad \text{(AdaMinSE)} \\
& d_i \leq e^{\eta} b_i + s_i, \quad \sum_{i \in [K]} s_i \leq \tau, e^{\eta} \alpha' + \tau \leq \alpha
\end{aligned}
$$

*Let $\mathbb{P}\left\{ Y \in C_i^{\alpha'}(X) \right\} \geq 1 - \alpha'$, for all $i \in [K]$. Moreover, consider the selection algorithm $\hat{S}(\xi, \varepsilon)$ with $\mathbb{P}\{\hat{S}(\xi, \varepsilon) = i | \xi\} = d^{\star}(b, \xi)$. Then, $\mathbb{P}\left\{ Y \in C_{\hat{S}(\xi, \varepsilon)}^{\alpha}(X) \right\} \geq 1 - \alpha$.*

*Proof.* The result can be recovered as follows

$$
\begin{aligned}
\mathbb{P}\left\{ Y \notin C_{\hat{S}(\xi, \varepsilon)}^{\alpha}(X) \right\} &= \mathbb{E}\left[ \sum_{i=1}^{K} d^{\star}(b, \xi) \mathbb{1}\left\{ Y \notin C_i^{\alpha'}(X) \right\} \right] \\
&\leq \sum_{i=1}^{K} e^{\eta} b_i \mathbb{P}\left\{ Y \notin C_i^{\alpha'}(X) \right\} + \tau \leq e^{\eta} \alpha' + \tau \leq \alpha,
\end{aligned}
$$

where the first and last inequalities follow from the two constraint $d_i \leq e^{\eta} b_i + s_i$ and $e^{\eta} \alpha' + \tau \leq \alpha$ in AdaMinSE, respectively. $\qquad \square$

Proposition 3 ensures that AdaMinSE achieves the desired coverage. By optimizing over $(\eta, \tau)$, it removes the need for manual tuning, making it a practical and reliable approach for selection.

## A.2 Derandomizing the Prediction Set

One potential limitation of the stable selection algorithms presented in Section 3.2 is that they produce a random confidence set, which may be undesirable in certain applications. In such cases, the stable selection process can be derandomized using techniques from [26]. This is formalized in the following proposition.

**Proposition 4** (Derandomized smallest conformal set). *Let $\hat{S}(\xi, \varepsilon)$ be an $(\eta, \tau)$-stable selection algorithm, and define $p_i(\xi) := \mathbb{P}\left\{\hat{S}(\xi, \varepsilon) = i | \xi\right\}$. Then, consider the derandomized confidence set*

$$C_{\mathrm{dr}}(X, \xi) := \left\{y \in \mathcal{Y} : \sum_{i=1}^{K} p_i(\xi) \mathbb{1}\left\{y \in C_i^\alpha(X)\right\} \geq \frac{1}{2}\right\}.$$

*If $C^\alpha(\cdot)_1, \ldots, C^\alpha(\cdot)_K$ satisfy (1), it holds that $\mathbb{P}\{Y \in C_{\mathrm{dr}}(X, \xi)\} \geq 1 - 2(\alpha e^\eta + \tau)$.*

*Proof.* The proof builds upon the reasoning in [26]. Since $\hat{S}(\xi, \varepsilon)$ is $(\eta, \tau)$-stability, we know that there exists a fixed point $b \in \Delta^{K-1}$ such that $p_i(\xi) \leq \exp(\eta) b_i + s_i$ and $\sum_{i=1}^{K} s_i \leq \tau$. It follows that

$$\mathbb{E}\left[\sum_{i=1}^{K} p_i(\xi) \mathbb{1}\left\{Y \notin C_i^\alpha(X)\right\}\right] \leq \exp(\eta) \alpha + \tau.$$

Moreover, using the Markov inequality, we have that

$$\mathbb{P}\{Y \notin C_{\mathrm{dr}}(X, \xi)\} = \mathbb{P}\left\{\sum_{i=1}^{K} p_i(\xi) \mathbb{1}\left\{Y \notin C_i^\alpha(X)\right\} \geq \frac{1}{2}\right\} \leq 2 \mathbb{E}\left[\sum_{i=1}^{K} p_i(\xi) \mathbb{1}\left\{Y \notin C_i^\alpha(X)\right\}\right],$$

from which the result follows immediately. $\square$

We highlight that while the set $C_{\mathrm{dr}}(\cdot, \xi)$ is still random with respect to the randomness of $\{C_i^\alpha(\cdot)\}_{i=1}^{K}$, the derandomization here refers to the fact that the stable selection process does not introduce additional randomness due to $\varepsilon$.

## A.3 Conditional Coverage

A stronger guarantee than marginal coverage (1) is conditional coverage. Let $G : \mathcal{X} \to \mathcal{G}$ be a function that maps an input $X$ to a group attribute $G(X) \in \mathcal{G}$. A conformal predictor $C(X)$ is said to satisfy $(1 - \alpha)$ conditional coverage with respect to $G$ if,

$$\mathbb{P}\{Y \in C(X)|G(X)\} \geq 1 - \alpha. \tag{10}$$

This ensures that the coverage guarantee holds not just on average over all $X$, but also when restricted to specific subpopulations defined by $G$. An example is the case where $\mathcal{G}$ is finite, which then corresponds to Group-Conditional validity [33]. Such guarantees are crucial for added reliability in many applications.

Our stability-based selection framework can be extended to preserve conditional coverage. Suppose we have $K$ conformal predictors $\{C_i^\alpha(X)\}_{i=1}^{K}$, each satisfying $(1 - \alpha)$ conditional coverage with respect to $G$. That is, for each $i \in [K]$,

$$\mathbb{P}\{Y \in C_i^\alpha(X)|G(X)\} \geq 1 - \alpha. \tag{11}$$

We can now state the conditional coverage guarantee for the selected set.

**Proposition 5** (Conditionally valid stable selection). *Assume that each conformal predictor $C_i^\alpha(X)$ satisfies $(1 - \alpha)$ conditional coverage with respect to $G$, for all $i \in [K]$. If $\hat{S} : \Xi \times \mathcal{E} \to [K]$ is an $(\eta, \tau)$-stable selection algorithm (with $\xi = [\lambda(C_1^\alpha(X)), \ldots, \lambda(C_K^\alpha(X))]$), then almost surely*

$$\mathbb{P}\left\{Y \in C_{\hat{S}(\xi, \varepsilon)}^\alpha(X) \mid G(X)\right\} \geq 1 - (\alpha e^\eta + \tau). \tag{12}$$

*Proof.* We want to bound the conditional miscoverage probability

$$P_{\text{miscover}|G(X)} := \mathbb{P}\left\{Y \notin C^{\alpha}_{\hat{S}(\xi(X),\varepsilon)}(X) \mid G(X)\right\}.$$

By the law of total expectation, conditioning on $X$ we have

$$P_{\text{miscover}|G(X)} = \mathbb{E}_{X|G(X)}\left[\mathbb{P}\left\{Y \notin C^{\alpha}_{\hat{S}(\xi(X),\varepsilon)}(X) \mid X, G(X)\right\}\right].$$

Since $G(X)$ is determined by $X$, the inner probability is $\mathbb{P}\left\{Y \notin C^{\alpha}_{\hat{S}(\xi(X),\varepsilon)}(X) \mid X\right\}$. Since the randomness $\varepsilon$ in $\hat{S}(\xi(X),\varepsilon)$ is independent of $Y$ given $X$, we have

$$\mathbb{P}\left\{Y \notin C^{\alpha}_{\hat{S}(\xi(X),\varepsilon)}(X) \mid X\right\} = \sum_{s=1}^{K} \mathbb{P}\left\{Y \notin C^{\alpha}_s(X) \mid X\right\} \mathbb{P}_{\varepsilon}\left\{\hat{S}(\xi(X),\varepsilon) = s \mid X\right\}.$$

Using $(\eta,\tau)$-stability (conditional on $\xi(X)$) property of $\hat{S}$, we have

$$\sum_{s=1}^{K} \mathbb{P}\left\{Y \notin C^{\alpha}_s(X) \mid X\right\} \mathbb{P}_{\varepsilon}\left\{\hat{S}(\xi(X),\varepsilon) = s \mid X\right\} \le e^{\eta} \sum_{s=1}^{K} \mathbb{P}\left\{Y \notin C^{\alpha}_s(X) \mid X\right\} \mathbb{P}\left\{S_0 = s \mid X\right\} + \tau$$

$$\le e^{\eta} \sum_{s=1}^{K} \mathbb{P}\left\{Y \notin C^{\alpha}_s(X) \mid X\right\} \mathbb{P}\left\{S_0 = s\right\} + \tau.$$

for some random $S_0$ variable with support $[K]$. It follows that

$$P_{\text{miscover}|G(X)} = e^{\eta} \sum_{i=1}^{K} \mathbb{P}\left\{S_0 = i\right\} \mathbb{E}_{X|G(X)}\left[\mathbb{P}\left\{Y \notin C^{\alpha}_i(X)\right\} \mid X\right] + \tau$$

$$\le e^{\eta}\alpha + \tau.$$

$\square$

This result shows that if the original conformal predictors provide conditional coverage, the stable selection mechanism allows for selecting among them while retaining a (correspondingly adjusted) conditional coverage guarantee. The required level for the initial predictors would be $1 - (\alpha - \tau)e^{-\eta}$ to achieve $1 - \alpha$ conditional coverage post-selection.

## B   Proofs

### B.1   Proofs of Section 3

The proof of Theorem 1 requires the following lemma from Zrnic and Jordan [4], which is inspired by the approach in [34].

**Lemma 4.** *[4, Lemma 1] Let $\hat{S} : \Xi \times \mathcal{E} \to \mathcal{S}$ be an $(\eta,\tau,\nu)$-stable selection algorithm and $S_0$ be the corresponding random variable. Then, $(\xi, \hat{S}(\xi,\varepsilon)) \approx_{\eta,\tau+\nu} (\xi, S_0)$.*

*Proof of Theorem 1.* In this proof, similar to analogous results in Zrnic and Jordan [4], a lot of the heavy lifting is done by Lemma 4. Nonetheless, the selection dependence on the data on the confidence sets themselves introduces some subtleties, making direct application of the steps of Zrnic and Jordan [4] non-straightforward. Thus, we take a different starting step by defining a shadow algorithm.

Let us define $\hat{S}' : \Xi \times \mathcal{Z} \times \mathcal{E} \to \mathcal{S}$ such that $\hat{S}'(\xi,\zeta,\varepsilon) = \hat{S}(\xi,\varepsilon)$, for all $(\xi,\zeta) \in \Xi \times \mathcal{Z}$. Then, if $\hat{S}$ is $(\eta,\tau,\nu)$-stable with respect to $\mathcal{P}$ on $\Xi$, we also have that $\hat{S}'$ is $(\eta,\tau,\nu)$-stable with respect to the product distribution $\mathcal{P} \otimes \mathcal{Q}$, where $\mathcal{Q}$ is the distribution of $\zeta$. Combining this with Lemma 4, we obtain

$(\zeta, \xi, \hat{S}(\xi, \varepsilon)) \approx_{\eta, \tau + \nu} (\zeta, \xi, S_0)$. Now, defining the event $O_\delta := \{(\zeta, \xi, \hat{S}(\xi, \varepsilon)) \in \mathcal{Z} \times \Xi \times \mathcal{S} : \zeta \notin \mathrm{CI}_{\hat{S}(\xi, \varepsilon)}^{\delta e^{-\eta}}\}$, we have that

$$\mathbb{P}\left\{\zeta \notin \mathrm{CI}_{\hat{S}(\xi, \varepsilon)}^{(\delta e^{-\eta})}\right\} \leq e^\eta \mathbb{P}\left\{(\zeta, \xi, S_0) \in O_\delta\right\} + \tau + \nu$$
$$= e^\eta \mathbb{P}\left\{\zeta \notin \mathrm{CI}_{S_0}^{\delta e^{-\eta}}\right\} + \tau + \nu \leq e^\eta \delta e^{-\eta} + \tau + \nu \leq \delta + \tau + \nu,$$

where the first inequality follows from the definition of indistinguishability, and the second inequality follows from the assumption that $\mathbb{P}\{\zeta \notin \mathrm{CI}_s^\alpha\} \leq \alpha$ holds for all $s \in \mathcal{S}$. $\qquad \square$

*Proof of Corollary 1.* The result follows immediately from Theorem 1 with $Y = \zeta$ and $\nu = 0$. $\quad \square$

*Proof of Lemma 1.* Let $S_0 = \arg\min_{i \in [K]} \varepsilon_i$, with $\varepsilon_i \overset{\text{i.i.d.}}{\sim} \mathrm{Lap}\,(1/\eta)$. Moreover, note that $\hat{S}(\xi, \varepsilon) = i$ if and only if $\varepsilon_i \leq \min_{j \neq i}\{\varepsilon_j + \lambda\left(\mathrm{C}_j^\alpha(X)\right) - \lambda\left(\mathrm{C}_i^\alpha(X)\right)\} \leq \min_{j \neq i} \varepsilon_j + 1$, where we used the fact that $\lambda(C_i^\alpha(X)) \in [0, 1]$. Finally, for all $i \in [K]$

$$\mathbb{P}\left\{\hat{S}(\xi, \varepsilon) = i \,\Big|\, \sigma(\xi, \{\varepsilon_j\}_{j \neq i})\right\} \leq \mathbb{P}\left\{\varepsilon_i \leq \min_{j \neq i} \varepsilon_j + 1 | \sigma(\xi, \{\varepsilon_j\}_{j \neq i})\right\}$$
$$\leq e^\eta \mathbb{P}\left\{\varepsilon_i \leq \min_{j \neq i} \varepsilon_j | \sigma(\xi, \{\varepsilon_j\}_{j \neq i})\right\}$$
$$= e^\eta \mathbb{P}\left\{S_0 = i | \sigma(\xi, \{\varepsilon_j\}_{j \neq i})\right\},$$

where the second inequality uses the fact that the densities ratio $p_{\varepsilon_i - 1}/p_{\varepsilon_i}$ is upper bounded by $e^\eta$. The result now follows by taking expectation on both sides. $\qquad \square$

*Proof of Lemma 2.* Let $S_0$ be a uniform r.v. with $\mathbb{P}\{S_0 = i\} = 1/k$. Then,

$$\frac{\mathbb{P}\left\{\hat{S}(\xi, \varepsilon) = i \,\Big|\, \xi\right\}}{\mathbb{P}\left\{S_0 = i\right\}} = \frac{k \exp\left(-\eta\lambda\left(\mathrm{C}_i^\alpha(X)\right)\right)}{\sum_{i \in [K]} \exp\left(-\eta\lambda\left(\mathrm{C}_i^\alpha(X)\right)\right)}$$
$$\leq \frac{k \exp\left(\eta\right)}{k \exp\left(-\eta\right)} = \exp 2\eta,$$

where we used the fact that $\lambda(C_i^\alpha(X)) \in [0, 1]$. $\qquad \square$

*Proof of Lemma 3.* The optimal solution $p^\star(b, \xi)$ satisfies $p_i^\star(b, \xi) \leq e^\eta b_i + s_i$, with $\sum_{i=1}^K s_i \leq \tau$, for all $i \in k$. Therefore, letting $S_0$ be a r.v. with $\mathbb{P}\{S_0 = i\} = b_i$. Then, for all $\mathcal{S} \subseteq [K]$, we have

$$\mathbb{P}\left\{\hat{S}(\xi, \varepsilon) \in \mathcal{S}\right\} \leq e^\eta \mathbb{P}\left\{S_0 \in \mathcal{S}\right\} + \tau,$$

which concludes the proof. $\qquad \square$

*Proof of Proposition 1.* By the assumption that $\mathcal{A}$ is $(\eta, \tau)$-stable w.r.t. $\mathcal{P}$, Definition 2 guarantees the existence of a r.v. $S_0$ such that $\mathcal{A}(\xi, \varepsilon) \approx_{\eta, \tau}^{|\xi} S_0$ holds $\mathcal{P}$-almost surely. This implies that for any $G \subseteq [K]$, $\mathbb{P}_\varepsilon\{\mathcal{A}(\xi, \varepsilon) \in G | \xi\} \leq e^\eta \mathbb{P}\{S_0 \in G\} + \tau$ holds $\mathcal{P}$-almost surely.

We set the prior vector $b \in \Delta^{K-1}$ as the distribution of $S_0$ by choosing $b_i := \mathbb{P}\{S_0 = i\}$. Let $p_i^\mathcal{A}(\xi) = \mathbb{P}_\varepsilon\{\mathcal{A}(\xi, \varepsilon) = i | \xi\}$. Define $s_i(\xi) = \max\left(0, p_i^\mathcal{A}(\xi) - e^\eta b_i\right)$. Clearly, $s_i(\xi) \geq 0$ and $p_i^\mathcal{A}(\xi) \leq e^\eta b_i + s_i(\xi)$ for all $i$. Let $\mathcal{S}^+(\xi) = \{i \mid p_i^\mathcal{A}(\xi) > e^\eta b_i\}$. Then, $\mathcal{P}$-almost surely,

$$\sum_{i=1}^K s_i(\xi) = \sum_{i \in \mathcal{S}^+(\xi)} (p_i^\mathcal{A}(\xi) - e^\eta b_i) = \mathbb{P}_\varepsilon\left\{\mathcal{A}(\xi, \varepsilon) \in \mathcal{S}^+(\xi) | \xi\right\} - e^\eta \mathbb{P}\left\{S_0 \in \mathcal{S}^+(\xi)\right\}$$
$$\leq (e^\eta \mathbb{P}\left\{S_0 \in \mathcal{S}^+(\xi)\right\} + \tau) - e^\eta \mathbb{P}\left\{S_0 \in \mathcal{S}^+(\xi)\right\} = \tau.$$

Since $p^\mathcal{A}(\xi)$ is a probability distribution, $p^\mathcal{A}(\xi) \in \Delta^{K-1}$. Thus, $p^\mathcal{A}(\xi)$ satisfies the constraints of the MinSE linear program with prior $b$ for $\mathcal{P}$-almost surely all $\xi$.

The MinSE algorithm finds the solution $p^\star(b, \xi)$ that minimizes the objective function $\sum_{i=1}^{K} p_i \lambda\left(C_i^\alpha(X)\right)$ over the set of all feasible distributions satisfying these constraints. Since $p^{\mathcal{A}}(\xi)$ is a feasible solution $\mathcal{P}$-almost surely, its objective value must be greater than or equal to the minimum objective value achieved by the optimal solution $p^\star(b, \xi)$. Therefore, $\mathcal{P}$-almost surely,

$$\sum_{i=1}^{K} p_i^\star(b, \xi) \lambda\left(C_i^\alpha(X)\right) \leq \sum_{i=1}^{K} p_i^{\mathcal{A}}(\xi) \lambda\left(C_i^\alpha(X)\right).$$

This completes the proof. □

## B.2 Proofs of Section 4

*Proof of Corollary 2.* Since $\hat{S}(\xi_t, \varepsilon_t)$ is $(\eta, \tau)$-stability. Using the same starting derivation as the proof of Proposition 1, we know that there exists a fixed point $b \in \Delta^{K-1}$ such that $p_i(\xi_t) \leq \exp(\eta) b_i + s_i$ and $\sum_{i=1}^{K} s_i \leq \tau$. Thus,

$$\mathbb{P}\left\{Y_t \notin C_{\hat{S}(\xi_t, \varepsilon_t)}^{(t)}(X_t)\right\} = \mathbb{E}\left[\mathbb{1}\left\{Y_t \notin C_{\hat{S}(\xi_t, \varepsilon_t)}^{(t)}(X_t)\right\}\right] = \sum_{i=1}^{K} p_i(\xi_t) \mathbb{1}\left\{Y_t \notin C_i^{(t)}(X_t)\right\}$$

$$\leq \sum_{i=1}^{K} e^\eta b_i \mathbb{1}\left\{Y_t \notin C_i^{(t)}(X_t)\right\} + \tau,$$

from which we have that

$$\limsup_{T \to \infty} \frac{1}{T} \sum_{t=1}^{T} \mathbb{P}\left\{Y_t \notin C_{\hat{S}(\xi_t, \varepsilon_t)}^{(t)}(X_t)\right\} \leq \limsup_{T \to \infty} \frac{1}{T} \sum_{t=1}^{T} \left(\sum_{i=1}^{K} e^\eta b_i \mathbb{1}\left\{Y_t \notin C_i^{(t)}(X_t)\right\} + \tau\right)$$

$$\leq e^\eta \sum_{i=1}^{K} b_i \left(\limsup_{T \to \infty} \frac{1}{T} \sum_{t=1}^{T} \mathbb{1}\left\{Y_t \notin C_i^{(t)}(X_t)\right\}\right) + \tau$$

$$\leq e^\eta \alpha + \tau,$$

where the second inequality follows from Fubini's theorem, which allows us to interchange the two sums, and the subadditivity of $\limsup$. This concludes the proof. □

*Proof of Proposition 2.* Using the definition of $C_{\text{comb}}^{(t)}$ in Algorithm 1, together with Markov's inequality, we have that

$$\mathbb{P}\left\{Y_t \notin C_{\text{comb}}^{(t)}\right\} = \mathbb{P}\left\{\sum_{i=1}^{K} p_i^\star(w^{(t)}, \xi_t) \mathbb{1}\left\{Y_t \notin C_i^{(t)}(X_t)\right\} \geq \frac{1}{2}\right\}$$

$$\leq 2 \mathbb{E}\left[\sum_{i=1}^{K} p_i^\star(w^{(t)}, \xi_t) \mathbb{1}\left\{Y_t \notin C_i^{(t)}(X_t)\right\}\right].$$

Moreover, from MinSE, we know that $p_i^\star(w^{(t)}, \xi_t) \leq e^\eta w_i^{(t)} + s_i$ and $\sum_{i=1}^{K} s_i \leq \tau$. Therefore,

$$\mathbb{P}\left\{Y_t \notin C_{\text{comb}}^{(t)}\right\} \leq 2 \left(e^\eta \mathbb{E}\left[\sum_{i=1}^{K} w^{(t)} \mathbb{1}\left\{Y_t \notin C_i^{(t)}(X_t)\right\}\right] + \tau\right) = 2\left(e^\eta \beta_t + \tau\right),$$

where the equality follows from (8). This concludes the proof.

□

## B.3 Proofs of Section 5

**Notation reminder.** Before providing the proof of Theorem 2, we introduce some additional notation and recall some notation in the main text. We consider a calibration dataset $\mathcal{D}_{\text{cal}} = \left\{(X_i, Y_i)_{i \in [m]}\right\}$ and a test point $(X, Y)$. We interchangeably denote $(X, Y)$ as $(X_{m+1}, Y_{m+1})$ and

define $\mathcal{D}_{\text{cal}}^+ = \left\{ (X_i, Y_i)_{i \in [m+1]} \right\}$. In addition, for each base predictor $k \in [K]$, we compute the non-conformity scores on the calibration data, for $i \in [m]$,

$$s_{k,i} := s_k(X_i, Y_i, f_k)$$

and denote the score of the test point as $s_{k,m+1} = s_k(X_{m+1}, Y_{m+1}, f_k)$. Let $T_k := \{s_{k,1}, \ldots, s_{k,m}\}$ denote the set of these calibration scores for model $k$. We also define $T_k^+ := T_k \cup \{s_{k,m+1}\}$. We use $s_{k,(r)}$ and $s_{k,(r)}^+$ to denote the $r$-th order statistic in $T_k$ and $T_k^+$. Furthermore, we denote the ranks of a score $s_{k,i}$ within the two sets as follows $R_{k,i} := \sum_{j=1}^m \mathbb{1}\{s_{k,j} \leq s_{k,i}\}$ and $R_{k,i}^+ := \sum_{j=1}^{m+1} \mathbb{1}\{s_{k,j} \leq s_{k,i}\}$. Similarly to Section 5, we parameterize the conformal prediction sets using ranks. For a rank index $R \in [m]$, define, respectively:

$$C_k(X, R) := \left\{ y \in \mathcal{Y} : s_k(X, y, f_k) \leq s_{k,(R)} \right\},$$
$$C_k^+(X, R) := \left\{ y \in \mathcal{Y} : s_k(X, y, f_k) \leq s_{k,(R)}^+ \right\}.$$

Both set families are monotonic: $C_k(X, R_1) \subseteq C_k(X, R_2)$ if $R_1 \leq R_2$; similarly for $C_k^+$. The relationship $s_{k,(R)} \geq s_{k,(R)}^+$ holds for $R \in [m]$, implying the set inclusion $C_k^+(X, R) \subseteq C_k(X, R)$. To relate with the standard perspective on split conformal threshold, note that for $i' \in [m]$

$$\mathbb{P}\left\{ R_{k,m+1}^+ \leq i' \right\} = \mathbb{P}\left\{ s_{k,m+1}^+ \leq s_{k,(i')}^+ \right\} \leq \mathbb{P}\left\{ s_{k,m+1}^+ \leq s_{k,(i')} \right\} = \mathbb{P}\left\{ Y_{m+1} \in C_k(X_{m+1}, i') \right\},$$

where $\mathbb{P}\left\{ R_{k,m+1}^+ \leq i' \right\} \geq i'/(m+1)$ by exchangeability. Note that this is equivalent up to a reparameterization to the split conformal method introduced in Section 2, by setting $i' = \lceil (1-\alpha)(m+1) \rceil$.

For each point $i \in [m+1]$, we apply the selection rule using its feature vector $X_i$ and independent randomness $\varepsilon_i \sim \mathcal{P}_\varepsilon$. Let $k_i := \hat{S}(X_i, \varepsilon_i)$ be the index of the predictor selected for the $i$-th data point. By our assumption, $k_i$ is independent of $\mathcal{D}_{\text{cal}}$, conditionally to $X_i$. We now define the **effective rank** for the $i$-th point. This is the rank of the $i$-th point's score calculated using the predictor $k_i$ that was selected specifically for $X_i$:

$$\hat{R}_i^+ := R_{k_i,i}^+ = R_{\hat{S}(X_i, \varepsilon_i),i}^+.$$

Similarly, for $i \in [m]$, we define

$$\hat{R}_i := R_{k_i,i} = R_{\hat{S}(X_i, \varepsilon_i),i}.$$

Finally, we define the following two sequences $\mathcal{R}^+ := (\hat{R}_1^+, \ldots, \hat{R}_{m+1}^+)$ and $\mathcal{R} := (\hat{R}_1, \ldots, \hat{R}_m)$, with $\hat{R}_{(i)}^+$ and $\hat{R}_{(i')}$ denoting their $i$-th and $i'$-th order statistics respectively, for $i \in [m+1]$ and $i' \in [m]$.

*Proof of Theorem 2.* Notice that $\mathcal{R}^+$ forms an exchangeable sequence [3, Lemma 2.2]. This follows from the exchangeability of the original data pairs $\{(X_i, Y_i)\}_{i=1}^{m+1}$ and the fact that the procedure to obtain $\hat{R}_i^+$ is symmetric w.r.t. $\mathcal{D}_{\text{cal}}^+$. In addition, $\mathcal{R} \subset \mathcal{R}^+$ implies that $\hat{R}_{(m)}^+ \leq \hat{R}_{(m)}$. As a direct consequence, for $i \in [m]$, we get:

$$\frac{i}{m+1} \leq \mathbb{P}\left\{ \hat{R}_{m+1}^+ \leq \hat{R}_{(i)}^+ \right\} \leq \mathbb{P}\left\{ s_{k_{m+1}, \left( R_{k_{m+1},m+1}^+ \right)}^+ \leq s_{k_{m+1}, \left( \hat{R}_{(i)}^+ \right)}^+ \right\}$$
$$= \mathbb{P}\left\{ s_{k_{m+1},m+1}^+ \leq s_{k_{m+1}, \left( \hat{R}_{(i)}^+ \right)}^+ \right\}$$
$$\leq \mathbb{P}\left\{ s_{k_{m+1},m+1}^+ \leq s_{k_{m+1}, \left( \hat{R}_{(i)} \right)} \right\}$$
$$= \mathbb{P}\left\{ Y_{m+1} \in C_{k_{m+1}} \left( X_{m+1}, \hat{R}_{(i)} \right) \right\}.$$

Here, the first line follows from the monotonicity of the order statistics of $\left(s^+_{k_{m+1},1}, \ldots, s^+_{k_{m+1},m+1}\right)$ and the definition of $\hat{R}^+_{m+1}$. The second line is by the fact that for any $k \in [K], i' \in [m+1]$, we have $s^+_{k,(R^+_{k,i'})} = s^+_{k,i'}$. The third by $\hat{R}^+_{(i)} \leq \hat{R}_{(i)}$ for any $i \in [m]$ and the fourth by the monotonicity of the split conformal set w.r.t. increasing score. Finally, choosing $i = \lceil (1-\alpha)(m+1) \rceil$, we get

$$\mathbb{P}\left\{Y_{m+1} \in C_{\hat{S}(X_{m+1})}(X_{m+1}, R_{(i)})\right\} = \mathbb{P}\left\{Y_{m+1} \in C_{k_{m+1}}(X_{m+1}, R_{(i)})\right\} \geq 1 - \alpha,$$

and recover the theorem. □

## C  Additional Experiments

### C.1  Batch Experiments

In Section 6, we provided some results in the batch setting. Here, using the same selection algorithms as in Section 6, we extend the experimental setting as detailed in the following subsections. We first provide additional information on the batch experiments.

For the real dataset experiments (Abalone, Bike Sharing [31, CC BY 4.0], California Housing [30, BSD License]), the hyperparameters of several base regression models were optimized prior to their use in the main conformal prediction experiments. This tuning was performed using "BayesSearchCV" from the `scikit-optimize` library [32, BSD-2 license], as mentioned in Section 6. The models subjected to this tuning process included RandomForestRegressor, GradientBoostingRegressor, ElasticNet, DecisionTreeRegressor, AdaBoostRegressor, and ExtraTreesRegressor. All experiments took approximately 200 CPU hours using 16 cores of Intel Xeon CPU Gold 6230 and 32 GB of system memory. For the implementation of ModSel and the structure of our code, we based our implementation on the version open-sourced by Liang et al. [11]. Nonetheless, we significantly deviated from their implementation to allow for more efficient parallel processing.

Both $X$ and $Y$ were standardized. The `BayesSearchCV` process was configured to run for 25 iterations (`n_iter=25`) with 3-fold cross-validation (`cv_folds=3`) for each model and hyperparameter setting. The nonconformity score function also utilized a RandomForestRegressor to predict residuals, and its hyperparameters were the same as the tuned parameters for RandomForestRegressor for prediction.

Finally, for the synthetic experiments to follow, we note that YK-Adjust produced infinitely large sets in some runs, as such it is not plotted.

#### C.1.1  Homogeneous vs Heterogeneous Data Preprocessing

In Section 6, we preprocessed the dataset, both synthetic and real, by splitting them to 5 disjoint equal subsets using constrained K-means [35, 36, Code under BSD 3-Clause License]. Here, we provide additional results, without this splitting step. We call the setting with no splitting, homogeneous, and the setting with splitting heterogeneous. The homogeneous setting may represent a more challenging environment for our approach and can be more favorable to the methods of Liang et al. [11] and Yang and Kuchibhotla [12]. In particular, in this homogeneous setting, one conformal predictor can typically be superior to all other predictors, as such simply selecting the best-on-average predictor can yield very competitive results, specially that the stability-based coverage guarantees can be conservative.

Using the same synthetic data setting as in Section 6, we report the homogeneous results in Figure 3. In addition, we report the homogeneous results on the real data experiments in Figure 4. For real data experiments, Recal stays competitive with baselines. Nonetheless, for the synthetic experiments, our approach performs worse than baselines.

#### C.1.2  Varying Data Generation in Synthetic Experiments

To further compare with baselines, we adjust the data generation procedure in the synthetic experiment to match one of the settings in [11]. In particular, we repeat exactly the same steps as the synthetic

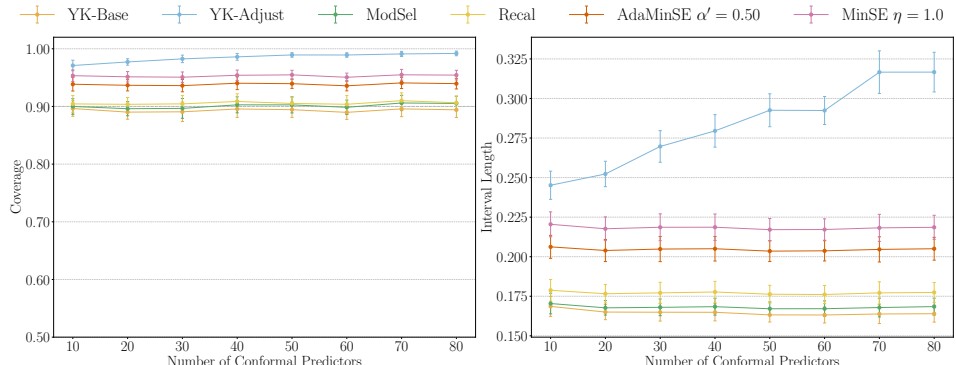

Figure 3: Homogeneous synthetic results. Each plot shows coverage (left) and interval length (right)

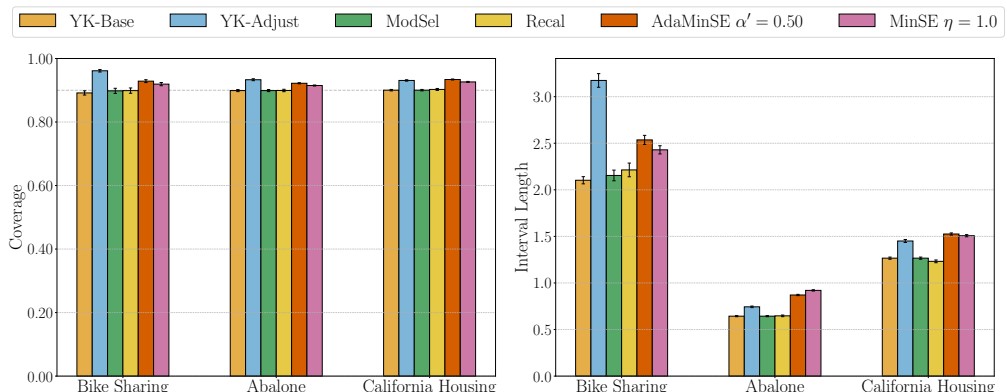

Figure 4: Comparison of UCI dataset results under homogeneous data processing.

experiments of Section 6, but replace the data generation procedure step by

$$X \sim \mathcal{N}(0, I_d), \quad \varepsilon \sim \mathcal{N}(0, 1)$$
$$\theta_i = \mathbb{1}\{i \mod 20 = 0\}$$
$$Y = X^T\theta + \varepsilon,$$

where $d = 300$. This matches the sparse normal setting with Gaussian noise in Liang et al. [11]. Given the linear dependency, this setting behaves more similarly to the homogeneous experiments; thus the globally optimal predictor may be inferred by predictors learning on any subset of the data. We report the results in Figure 5. We observe that Recal achieves very similar results to ModSel. YK-Base produces smaller conformal sets at the cost of higher miscoverage.

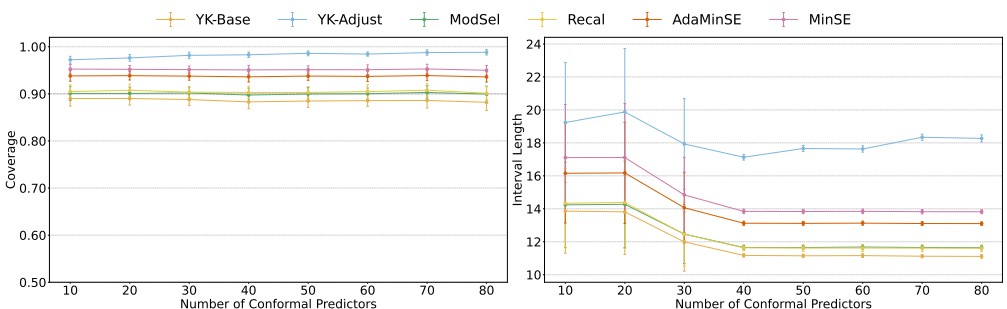

Figure 5: Comparison with baselines in the sparse linear setting with Gaussian noise

### C.1.3 Comparison against Average Model Baseline

Furthermore, we repeat the experiments of synthetic experiments of Section 6, but add an additional baseline. In particular, we consider the average model baseline, which averages all predictors trained on the data and conformalizes the average predictor. This baseline may be interpreted as a simple model averaging approach. We report the results in Figure 6, denoting this additional baseline as AvgSplit. We observe that Recal, AdaMinSE, and MinSE perform better than the average model baseline.

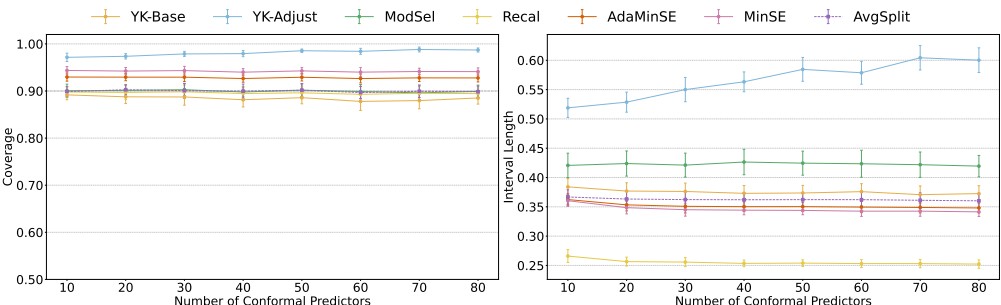

Figure 6: Comparison with average model as a baseline

### C.1.4 Effect of the Number of Calibration Points

In addition, we repeat the experiments on synthetic data, while varying the number datapoints in the calibration dataset. This mainly affects the results of YK-Adjust as its performance improves with larger calibration datasets. We report the corresponding results in Figure 7 and Figure 8.

### C.1.5 Additional Classification Experiments

We complement our regression studies with a compact ImageNet-1k classification [37, Non-Commercial Use] experiment intended to emulate a heterogeneous setting. Concretely, we construct two ViT-Base models from the same pretrained model: one kept *clean*, and one *degraded* by randomly shuffling the top-$k$ logits on a fraction of examples (we use a moderate corruption fraction of $10\%$ and $k = 20$). Then, each of the two models is used to construct a conformal predictor. For the models, we used the pretrained `timm` checkpoints [38].

Then, we construct 10 conformal predictors by mixing the two base conformal models in different ways, i.e. each model outputs the predictions of the *clean* for $50\%$ samples and *degraded* for the remaining. We use this structure to where different predictors perform better/worst on different samples.

Using this construction, we compare YK-Base, MinSE, and AdaMinSE, which operate purely at the set level – inspecting only the resulting conformal sets without requiring access to the underlying scores or split-conformal internals. We also evaluate a score-averaging baseline of Luo and Zhou [13] that linearly combines per-class scores via simplex weights learned on held-out indices. Because this baseline requires score access, we perform the mixing at the logit/score level rather than at the set level. We report the results in Figure 9. We note that this setting is one where the performance of the different predictors varies across the input space, which is precisely where our approach is most beneficial.

### C.2 Online Experiments:

**Online Setting Experiments**: we tested our online algorithm, AdaCOMA, by constructing an online analogue of our heterogeneous batch setting, where the performance of different predictors vary across time. The key advantage of AdaCOMA lies in its ability to condition selection on the current features $X_t$, via the observed interval sizes $\xi_t$ used in the stable selection mechanism. In contrast, COMA relies solely on historical performance in determining its weights $w^{(t)}$ (Algorithm 1).

To evaluate this, we designed an environment in which both COMA and AdaCOMA track and assign weights to $K = 10$ distinct forecasters. These forecasters are not fixed models; instead,

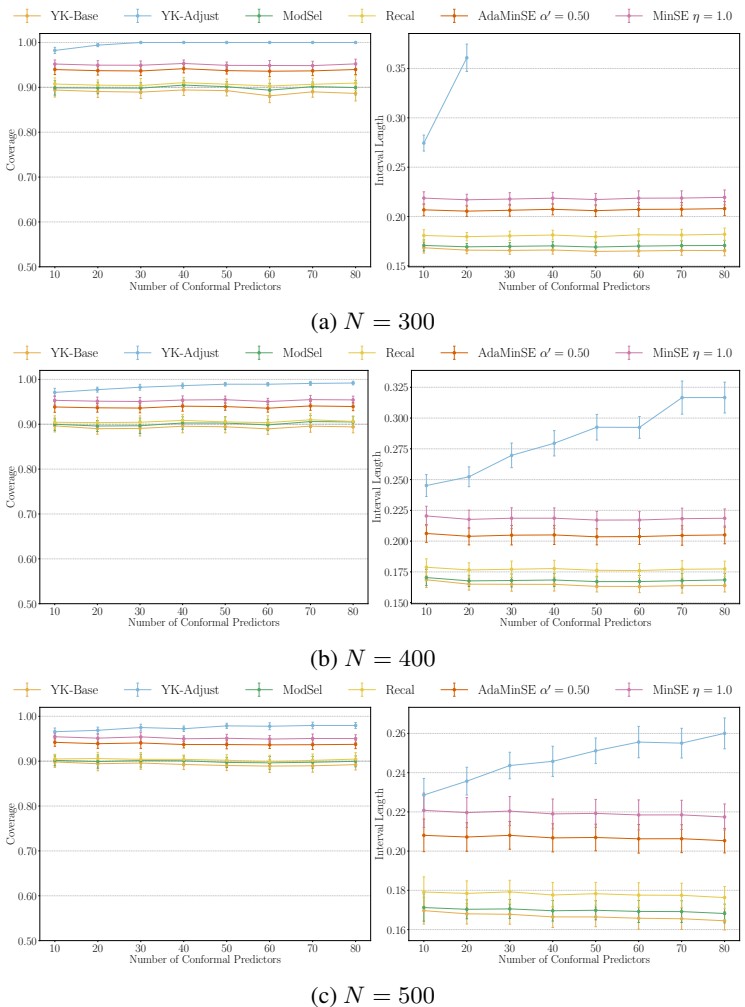

(a) $N = 300$

(b) $N = 400$

(c) $N = 500$

Figure 7: Homogeneous synthetic results. Each plot shows coverage (left) and interval length (right) for a different number of calibration examples ($N$).

they dynamically generate prediction intervals by drawing from a smaller pool of $M = 6$ diverse base online conformal algorithms. Each base algorithm is an instance of Adaptive Conformal Inference (ACI) [6] applied to an online learning model. For each of the $K$ forecasters, we simulate a heterogeneous environment where the optimal conformal predictor varies over time by partitioning the input data stream conceptually as follows: at the start of the experiment, each of the $K$ forecaster pick one model from the smaller subset of $M$ models. Then each $\tau = 50$ timesteps, the forecasters pick a different models, and so on. This setup, where forecasters change their models each $\tau$ timesteps, aims to simulate an environment, which requires the selection algorithm choosing among the forecasters to be strongly adaptive.

**Base Online Conformal Algorithms.** The $M = 6$ base algorithms were instances of Adaptive Conformal Inference (ACI) [6]. Default ACI parameters were: initialization period of 100 timesteps, we used an adaptive ACI stepsize adapted from the original code of [10]. The underlying online learning models were:

- Two SGD regressors: one with L1 penalty (Lasso, $\eta_0 = 0.001$, penalty $\alpha = 0.1$) and one with L2 penalty (Ridge, $\eta_0 = 0.001$, penalty $\alpha = 0.1$).
- Two SGD regressors (no penalty) with learning rates $\eta_0 \in \{0.001, 0.005\}$.
- Two Rolling Linear Regression models with window sizes of 50 (retrain frequency 12) and 100 (retrain frequency 25).

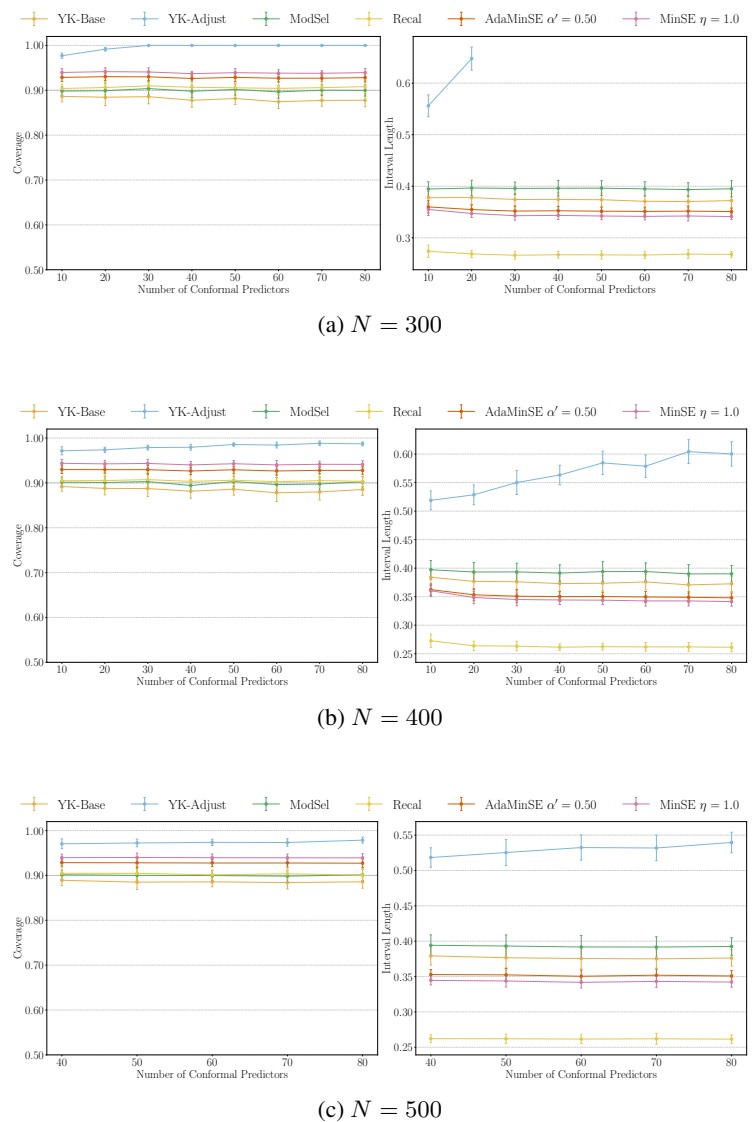

(a) $N = 300$

(b) $N = 400$

(c) $N = 500$

Figure 8: Heterogeneous synthetic results. Each plot shows coverage (left) and interval length (right) for a different number of calibration examples ($N$).

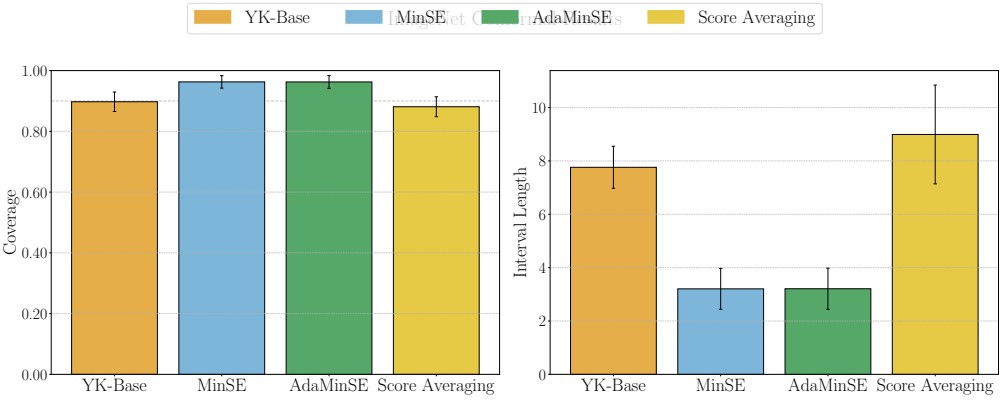

Figure 9: ImageNet classification results

We conducted experiments on two datasets, used for evaluation in Gasparin and Ramdas [10], ELEC [39, CC-BY 4.0] and a synthetic data generate according to the ARMA(1,1) model [40, Chapter 2]. For the precise data generation procedure for ARMA(1,1) model, refer to [10, Page 18.]. For both, AdaCOMA and COMA, we used two algorithms for the weights $w^{(t)}$ (Algorithm 1) over the forecasters: AdaHedge and Hedge with learning rate $\eta = 0.1$. We repeated the experiment for 50 seeds. For ARMA(1,1) dataset, 4 runs experienced numerical instability producing interval length multiple orders of magnitude above the rest for both COMA and AdaCOMA. We excluded those runs in calculating the reported results. The results are presented in Table 1. For COMA, we ran the underlying predictors using ACI with nominal miscoverage rate 0.1. For AdaCOMA, we ran ACI with nominal miscoverage rate of 0.09 and used AdaMinSE for the selection. For AdaMinSE selection, we tuned the selection such that COMA and AdaCOMA achieve similar coverage. The results are reported in Table 1. For ELEC dataset, AdaCOMA significantly outperformed COMA. For ARMA(1,1), both methods performed similarly with a small advantage to AdaCOMA.

Table 1: Comparison of COMA and AdaCOMA with different underlying aggregation algorithms (AdaHedge, Hedge) on ELEC and ARMA(1,1) datasets. Values are mean $\pm$ standard deviation (divide by the root of the number of seeds ($1/\sqrt{50}$) for the standard error). The target miscoverage is $\alpha = 0.1$.

| Dataset | Method | Avg. Miscoverage | Avg. Length |
|---|---|---|---|
| ELEC | COMA (AdaHedge) | $0.0942 \pm 0.002$ | $0.69 \pm 0.06$ |
| | AdaCOMA (AdaHedge) | $0.0959 \pm 0.001$ | $0.32 \pm 0.09$ |
| | COMA (Hedge) | $0.0942 \pm 0.002$ | $0.69 \pm 0.06$ |
| | AdaCOMA (Hedge) | $0.0963 \pm 0.001$ | $0.31 \pm 0.01$ |
| ARMA(1,1) | COMA (AdaHedge) | $0.102 \pm 0.001$ | $4.40 \pm 0.91$ |
| | AdaCOMA (AdaHedge) | $0.101 \pm 0.001$ | $4.23 \pm 0.65$ |
| | COMA (Hedge) | $0.102 \pm 0.001$ | $4.40 \pm 0.92$ |
| | AdaCOMA (Hedge) | $0.101 \pm 0.001$ | $4.31 \pm 1.06$ |

