# OpenReview forum: "Valid Selection among Conformal Sets"
_NeurIPS.cc/2025/Conference — NeurIPS 2025 poster_

### Official Review · Reviewer_bYvi · 2025-06-23

**Clarity:** 2
**Significance:** 2
**Originality:** 4
**Rating:** 4
**Confidence:** 3

**Summary:**

This paper studies the problem of constructing a conformal set when K different conformal predictors are given. Both the batch setting (i.e. standard split conformal prediction) and the online setting (without stationarity assumptions) are considered. A central point of departure is the notion of algorithmic stability, which allows for constructing valid conformal sets when an optimization step over the K existing conformal sets is needed (Theorem 1 and Corollary 1). In Lemma 3 a concrete algorithm that has theoretical coverage guarantees is derived. In Section 4 the online setting is considered, and a theoretical comparison with the existing COMA algorithm is made. In Section 5 the method is further improved by calibration after selection using effective ranks. This leads to Theorem 2, which shows that the method achieves marginal coverage after optimizing for the best among K conformal predictors. In Section 6 the new methods are experimentally compared to a few baselines on synthetic data and three real-world regression datasets.

**Questions:**

See weaknesses.

**Ethical Concerns:**

["NO or VERY MINOR ethics concerns only"]

**Final Justification:**

I think that the arguments given by the authors in favor of their approach compared to the mean ensemble as a baseline can make sense, but this all still has to be tested empirically. However, I completely understand this is not possible within a one week rebuttal period. In summary, I think that the method has interesting novel components, but I am not sure whether it performs as good as the authors claim. I will stick to my score of a "borderline accept".

**Limitations:**

Limitations of the presented methods are not discussed.

**Paper Formatting Concerns:**

I miss a conclusion.

**Quality:**

3

**Strengths And Weaknesses:**

For me the main strength of the paper is that it introduces quite some concepts that are new in the area of conformal prediction (according to my knowledge of the literature). These concepts can be useful when aggregating different conformal predictors in an ensemble learning setting. The authors clearly show that they have a strong knowledge of the theory in this field. The paper is written in a mathematically rigorous way.

The main weakness of the paper is that it is hard to follow the story (even for researchers with prior knowledge of conformal prediction I would say). The paper is written in a very dense manner, and many design choices are not really justified. I have read the paper two times. The first time I had a hard job to understand Section 3-6. The second time it was better, but I still have a lot of questions. Here I am listing the points that were not clear to me.

- It is not really discussed in which ML setting the proposed algorithms would be useful. I would guess that the main application domain is ensemble learning, either ensembles constructed using bootstrapping, or ensembles that involve randomizing model parameters (such as deep ensembles). Please be more specific in this.
- For such a setting, an obvious baselines would be performing conformal prediction on the mean of the ensemble. Why is this not a useful approach and inferior to the complicated aggregation that the authors propose?
- In the experiments on synthetic data, an ensemble is created by randomizing the kernel function and the regularization parameter in kernel ridge regression. Why would that be useful? I would argue that one should first optimize the hyperparameters using grid search, and run conformal prediction on the best performing model in terms of MSE. Would such an approach not perform better? Why not consider bootstrapping or another more useful technique to construct an ensemble?
- In the experiments the baselines that are implemented are references [11] and [12]. For [11] it is more or less clear to me what it does, but for [12] the authors only say that it is the "best-on-average" conformal predictor. I have no idea what this means without reading carefully [12]. More information is needed for readers that don't know [12].
- From Corollary 1 I conclude that the theoretical coverage highly depends on the hyperparameters \nu and \tau. In the experiments they are just set to one and zero. Isn't this like a very loose coverage bound? (Instead of 1-\alpha coverage one obtains 1-\alpha\epsilon coverage, which is much lower?
- In the introduction the authors write: "In contrast to [11] and [12] our method allow pointwise selection depending on each X". I don't see why this claim would be the case. The way I interpret the MinSE optimization problem in Lemma 3 is that the best convex combination of model is found. The same p_i values are used for all X, so the convex combination is not specific for each X?
- In the experiments on real data, what are the K models? Are these the six different regression models from scikit-learn, so K=6? Not 100% clear.
- The absolute residuals are considered as nonconformity scores, leading to prediction intervals that have the same length for all instances. This might be a clear disadvantage when interval size is a criterion, so it should be relatively easy to outperform baselines using this nonconformity score. I would be interested to see experiments where estimated quantiles are used as nonconformity scores (i.e. conformalized quantile regression). I expect that it becomes much more difficult to outperform the baselines using such a nonconformity score.
- Would the proposed method also be useful in classification settings? For such settings, specific nonconformity scores that optimize set size have been proposed, such as the adaptive prediction sets (APS) and its variants. A comparison to such methods could also be useful.

A lot of these questions are related to explaining the experiments better and comparing to different baselines. The experiments were in general a bit underwhelming, but I did not check the additional experiments in the appendix.

---

> ### Author Rebuttal · Authors · 2025-07-31
>
> We thank the reviewer for their detailed feedback and for recognizing our work's mathematical rigor and originality. We are glad you found our contributions noteworthy. Your comments on the paper's clarity and experimental justification are very helpful, and we address them below. They are roughly by the order of first mention of the review.
>
> 1. **On the paper's clarity and flow:**
>     We agree that the paper is dense and thank you for taking the time for a second reading. We will use the extra page for the camera-ready version to revise the manuscript, focusing on improving the flow and better motivating our design choices to make the story easier to follow.
>
> 2. **Application Setting and Ensembling:** Thank you for this question, which allows us to clarify a key aspect of our contribution. There are two ways to use ensembles for uncertainty quantification: (1) average the model predictions and then conformalize the final predictor, or (2) generate a conformal set for each model and then select among the sets. Our work focuses on the second, which can be seen as "ensembling in the space of conformal predictors." This is a general concept that applies even with a **single** underlying prediction model, which itself can be an ensemble. For instance, one could have a single model but multiple different nonconformity scores. Our framework provides a valid way to select the among the different conformal prediction strategies. As such, even with the mean of an ensemble, it is possible to leverage it to build multiple conformal predictors. Thus, even in this case, our approach can still be valuable to select among different conformal predictors.
>
>
> 3. **Experimental Motivation:** The synthetic experiment aimed to illustrate this core idea by creating a diverse pool of conformal predictors. To make the application setting clearer, we will add a new experiment in the revision where we fix the underlying regression model but vary the conformalization procedures themselves. As such, we will have different conformal predictor build on the same single model.
>
>
> 4. **On the baselines from references [11] and [12]:** The baseline from [12] is the *"best-on-average"* conformal predictor in the sense that all $K$ conformal predictors are reevaluated on the full calibration set. Then, a single predictor (among the $k$ predictors) with best average performance (e.g., smallest average set size) is identified and then use that predictor for all future test points. This contrasts sharply with our method, which makes a dynamic, **pointwise** selection for each new point. We are pointwise in the sense that for every new test point $X$, we evaluate the conformal sets at $X$, and try to select the smallest set at $X$. To refer to example 3 in the paper, a best-on-average predictor will try to find the better performing conformal predictor among the third and fourth images ($f(X)=X$ and $f(X)=-X$) based on the average set size. Then, this predictor will be used for all future test points. Meanwhile, a pointwise mechanism makes the choice based on the specific realization of $X$ and as such produces the fifth figure ($\eta=\ln(2)$).
>
> 5. **On the looseness of the coverage bound:** The bound involving the $e^\eta$ factor, while potentially conservative, is **tight**. We show in Example 1 that there exist pathological (but valid) settings where the miscoverage is exactly $e^\eta \alpha$. This is the theoretical price for a guarantee that holds under minimal assumptions, even in non-stationary online settings. For practical use, we propose two solutions that address this: `AdaMinSE`, which adaptively tunes the stability parameters, and our `Recal` procedure, which completely circumvents this factor in split-conformal settings by performing recalibration after the selection procedure.
>
> 6. **On the "pointwise selection" claim.**  This is an important point: the MinSE optimization is solved **separately for each test point $X$**. The optimization's input is the vector of set sizes $\xi = [\lambda(C_1(X)),\ldots, \lambda(C_k(X))]$, which explicitly depends on $X$. Consequently, the optimal probabilities $p_i(\xi)$ are a function of the current test point, enabling true pointwise selection. We will revise the text to emphasize this mechanism and clarify this aspect in the paper.
>
> 7. **Real data models:** Indeed, those are six regression models from scikit-learn and $K=6$.
>
> 8. **On the nonconformity score and interval length:** We do *not* use absolute residuals, which would lead to constant-length intervals. Our nonconformity score is $s_i(X, Y) = |f_i(X) - Y|/g_i(X)$, where $g_i(X)$ is a model trained to predict the residuals. This creates *adaptive* prediction intervals whose lengths vary with the input $X$, allowing each method to produce tighter intervals for "easier" instances.
>
> 9. **On the details of the real-data experiments.**
>     For the real-data experiments, we used $K=6$ models, and we will make this explicit.
>
> 10. **On the applicability to classification:** Yes, our method is directly applicable to classification. To demonstrate this, we conducted a new set of experiments on the ImageNet_1k, which we will include in the revised manuscript. We believe this new experiment provides a compelling use case for our proposed selection framework.
>
>     The experimental setup was designed to create a **heterogeneous** and challenging selection environment. We began with two pre-trained base models with different performance characteristics—one with a top-1 accuracy around 90% and another around 80%—and converted each into a standalone conformal predictor. These models naturally exhibit varying performance across the diverse classes in ImageNet. For the conformal scores, we used an adaptive score similar to the one used in the Appendix of [11].
>
>     From these two base predictors, we constructed *eight  composite predictors*. Each composite model creates a unique "risk profile" by outputing the conformal set either from the high-accuracy model  or from the lower-accuracy model, depending on the specific input $X$. These eight models use different probability distributions for this choice, creating a diverse pool of predictors for the selection algorithms to manage.
>
>    We compared the following selection methods `YK-Base`, `MinSE`, and `AdaMinSE`. The results, averaged over multiple 10 runs, are presented below.
>
> | Method   | Coverage (Target: 0.90) | Avg. Set Size |
> | :------- | :----------------------- | :------------ |
> | YK-Base  | 0.895                      | 3.93  |
> | MinSE    |  0.962     | 3.36  |
> | AdaMinSE | 0.949     | 3.27  |
>
> As shown, our proposed methods (`MinSE`, `AdaMinSE`) successfully achieve higher coverage with smaller sets.  Furthermore, their pointwise selection capability allows them to produce smaller average prediction sets than the non-pointwise `YK-Base` baseline, demonstrating their advantage in more heterogenous settings.
>
>
> * **On the missing Limitations section.* We tried to discuss limitation in the checklist and throughout the paper. Nonetheless, we agree that a dedicated limitations section is of value and we will add one to the revised manuscript.
>
>
>
> We thank the reviewer for his support and feedback. We remain available for any further discussion.
>
>
> ---
>
> **Erratum and recap on experiments**
> We wanted to openly disclose that we detected after submission a typo in the code, in file `batch_setting/rescale_residual.py` line 162, `cal_sigma` should be `cal_sigma[m]`. This changes the performance of ModSel, that is ultimately on-par with YK-base (yet never better, in fact YK-base lower bounds the confidence set size of ModSel). Other methods are unaffected. It thus does not change the quality of the "best competitor".
>
> This also doesn't change the conclusion of our experiments, summarized below:
> - on **bike sharing**, our methods MinSE and AdaMinSE  improve coverage by 2% while having a average length 10% below YK-Base. Recal achieves nominal coverage with set length more than 30% smaller.
> - on **abalone**, differences are more minor: both YK-base and ModelSel undercover by 1%,  Recal is valid and reduces set size by 5% against all competitors, while MinSE and AdaMinSE suffer from conservatism, overcovering by 2% and increasing set size by 10% w.r.t. YK-base
> - on **California housing**, YK-base, ModSel, MinSE, AdaMinSE achieve the same length, but with more than 2% of extra coverage for MinSE and AdaMinSE. Recal has nominal coverage with length 25% smaller than any other competitor.

---

> ### Comment · Area_Chair_VQCz · 2025-08-05
>
> Thanks for the insightful discussion.
>
> I am not an official reviewer for this paper. The following comment is not part of a formal review, and I want to avoid any undue influence on the discussion. My intent is simply to offer a pointer to the literature that might be relevant here. Please feel free to use or disregard this as you see fit.
>
> Given two score functions, $s_0(x,y)$ and $s_1(x,y)$, the method in YK et al. performs a selection between them. This is equivalent to choosing a weight $w \in \\{0,1\\}$ for a blended score function:
> \\[s_w(x,y) = (1-w) s_0(x,y) + w s_1(x,y).\\]
> A natural extension is to consider aggregation by allowing $w$ to be continuous in $[0,1]$ (e.g., $w=0.5$ for a simple average). This relates to work by Liang et al., where one could search for an optimal score by discretizing this range, e.g., finding the best score function among $s_0, s_{0.01}, s_{0.02}, \dots, s_{1}$.
>
> For a different perspective, some work explores directly averaging the scores. For example, in Luo \& Zhou (2025), given concavity of the score functions, the bound depends only on the dimension of $w$.
>
> Luo \& Zhou, "Conformity Score Averaging for Classification," https://openreview.net/pdf?id=Pvfd7NiUS6

---

> > ### Comment · Reviewer_bYvi · 2025-08-06
> > **Keep my score**
> >
> > I think that the arguments given by the authors in favor of their approach compared to the mean ensemble as a baseline can make sense, but this all still has to be tested empirically. However, I completely understand this is not possible within a one week rebuttal period. In summary, I think that the method has interesting novel components, but I am not sure whether it performs as good as the authors claim. I will stick to my score of a "borderline accept".

---

> > ### Author Response · Authors · 2025-08-08
> >
> > We are very grateful to the Area Chair for engaging with our discussion and for providing this relevant and helpful pointer. This work on score aggregation indeed complements our discussion of related methods by Liang et al. and Yang et al., and we will add it to our revision.
> >
> > Like the work of Liang et al. and Yang et al., the approach of Luo & Zhou aims to find an optimal global aggregation scheme. This provides a valuable contrast to our framework's focus on pointwise selection, and we believe that a direct comparison would be relevant.
> >
> > Implementing the approach of Luo & Zhou requires a classification setting and explicit access to the score functions. In contrast, our framework (with the exception of Recal) only requires black-box access to the final conformal sets. As the classification experiments added in the rebuttal process assumed only black-box access to the conformal algorithms, we will add a new set of classification experiments to the camera-ready version of our paper to enable a comparison with Luo & Zhou.
> >
> > We appreciate you providing this valuable context for our work.

---

### Official Review · Reviewer_6BVL · 2025-06-23

**Clarity:** 3
**Significance:** 3
**Originality:** 4
**Rating:** 5
**Confidence:** 4

**Summary:**

Assume k different conformal predictors (e.g. score taken from an ensemble of models). Directly picking the smallest set among the prediction sets, results in breaking the marginal guarantee in conformal prediction. Same applies to any selection that tunes some objective as it becomes data-dependent. The paper addresses this by valid stable selection. The authors show a set of selections that are stable, and they bound the coverage guarantee of a selection parameterized by the stability parameters. Furthermore they discuss the same setup for time-series data.

**Questions:**

1. What is the probability in Eq. 3 over?
2. Why the evaluation on real datasets are so limited? Why didn’t the authors also discuss classification. From what I understand application on classification setup is not a limit of their framework. The models here are not as complicated as a deep model.
3. If I understood correctly, your definition of stability is based on a reference (intuitively center) distribution. But can we also redefine this based on two random samples of the distribution. Intuitively I think if two samples are not too far with a high probability, they are also both close to a central reference sample.
4. What prevented authors to evaluate their result on a more complicated setup like ensemble image classification?
5. Also since the paper defines stable way to predict time-series data, why didn’t the authors try the algorithm on some time series task; e.g. stocks forecast, electricity demand forecast, etc.
6. I did not fully understood your definition of adversarial setup, and solution for it. To me adversarial setup is defined by a threat model including the set of possible perturbations, and the objective the adversary can achieve. Can you elaborate on this definition?

If my questions are answered and mentioned weaknesses are addressed, I’ll review the work again and try to increase my score.

**Ethical Concerns:**

["NO or VERY MINOR ethics concerns only"]

**Final Justification:**

I read the response by the authors. It addressed my comments. As mentioned while writing the review, I find the contribution considerable and the mathematical justifications to be solid.

**Limitations:**

No limitations mentioned. But I do not see specific limitations.

**Paper Formatting Concerns:**

I did not find a conclusions/limitation section which I am used to see in all scientific papers. I think it is essential for them to have it even if they an example to their appendix.

**Quality:**

3

**Strengths And Weaknesses:**

**Strength.**  I think the paper is mathematically solid, and also the contributions are noteworthy, despite their limited evaluations, I was convinced to accept the paper even around Section 3.3.

Also the paper addresses an interesting problem in CP. Basically they define a lower bound on the prediction set defined from an ensemble of CPs while it is data dependent tuning for a specific goal; e.g. smaller set size.

**Weakness: Empirical evaluation.**  I believe in contrast with the complicated theory, there could be evaluation done on larger datasets, and more complicated model. Of course, authors can easily examine their method on an ensemble of classifiers for the CIFAR10 dataset.

I suggest some rephrasing, however they are only for the readability; feel free to ignore them.

**Line 29.** It is better to reorder the words: In section 4, we extend our work …

**Line 115.** It makes it more understandable if you use an example for the inner-randomness of an algorithm. To me, one example is the uniform random value in the APS score function.

**Line 117.** I suggest to use another notation for algorithms that is dissimilar to random variable, unless there is an inherent similarity in the nature of both entities.

**Line 130.** The random algorithm is stable not stability. Right?

**Examples.** I think the authors have used a block similar to theorem for their examples. The readers eye gets tired as there is an italic text for several paragraphs. I suggest the authors to try some other type of environment that has font in normal style.

**No conclusion, and limitations.** Also why there is no conclusions section?

---

> ### Author Rebuttal · Authors · 2025-07-31
>
> We thank the reviewer for their constructive review. Your feedback is very valuable in improving our work. We address your questions and suggestions below.
>
>
> * **Regarding Empirical Evaluation:** We thank you for this constructive suggestion. Your feedback prompted us to run new experiments on an ImageNet-1k classification task to better demonstrate the generality and scalability of our framework. Our experimental setup was designed to create a *heterogeneous* and challenging selection environment. We began with two pre-trained base models with different performance characteristics—one with a top-1 accuracy around 90% and another around 80%—and converted each into a standalone conformal predictor. These models naturally exhibit varying performance across the diverse samples of ImageNet. For the conformal scores, we used an adaptive score similar to the one used in the Appendix of [11]. From these two base predictors, we constructed *eight  composite predictors*. Each composite model creates a unique "risk profile" by outputing the conformal set either from the high-accuracy model or from the lower-accuracy model, depending on the specific input $X$. These eight models use different probability distributions for this choice, creating a diverse pool of predictors for the selection algorithms to manage. All the composite predictors achieve marginal coverage guarantee.
>
> We compared the following selection methods `YK-Base`, `MinSE`, and `AdaMinSE`. The results, averaged over 10 runs, are presented below.
>
> | Method   | Coverage (Target: 0.90) | Avg. Set Size |
> | :------- | :----------------------- | :------------ |
> | YK-Base  | 0.895                      | 3.93  |
> | MinSE    |  0.962     | 3.36  |
> | AdaMinSE | 0.949     | 3.27  |
>
> As shown, our proposed methods (`MinSE`, `AdaMinSE`) successfully achieve higher coverage with smaller sets. This is due to their pointwise selection capability, which allows them to produce smaller average prediction sets than the non-pointwise `YK-Base` baseline.
>
>
> * **Time-Series Evaluation:** As you correctly noted, our method is well-suited for time-series data. We would like to highlight the online experiments already present in the appendix, where AdaCOMA algorithm is evaluated on electricity forecasting and a synthetic task.
>
> * **Regarding Readability,  Presentation, and Formatting:** Thank you for these helpful suggestions. We will incorporate all of them in the revised manuscript to improve clarity. We also believe that making the example non-italic is easier on the eye. Using the extra page allowed in the camera-ready version, we will a conclusion section, which discusses the limitation.
>
>
> * **Response to Other Questions:**
>
>     *  **What is the probability in Eq. (3) over:** The probability is taken over $\zeta$ and the randomness of CI itself. The statement holds for all $s$ and $\alpha$ is assumed to be fixed.
>     * **On distributional notion of stability:** This is an insightful question. Our definition of stability is asymmetric (bounding an output distribution relative to a fixed one) and it is the minimal condition required for our coverage proofs to hold. I.E., the stability statement is only required with the reference distribution on the right-hand side and the output distribution on the left-hand side, but not vice-versa. We believe it would be possible to use the notion of stability you proposed to get our coverage guarantees. Nonetheless, due to symmetry, it may be stronger than necessary for the proofs.
>
>     3.  **"adversarial" setup:** We use the term "adversarial" in the standard sense from online learning theory. It means the sequence of data points $(X_t, Y_t)$ can be chosen by an adversary, which allows us to prove worst-case robustness to arbitrary distribution shifts over time. This is distinct from the concept of adversarial examples (i.e., perturbed inputs). We will clarify this distinction in the text.
>
> We hope these responses and our new experiments fully address your concerns. We are happy to answer any further questions during the discussion period.
>
>
> ---
>
> **Erratum and recap on experiments**
> We wanted to openly disclose that we detected after submission a typo in the code, in file `batch_setting/rescale_residual.py` line 162, `cal_sigma` should be `cal_sigma[m]`. This changes the performance of ModSel, that is ultimately on-par with YK-base (yet never better, in fact YK-base lower bounds the confidence set size of ModSel). Other methods are unaffected. It thus does not change the quality of the "best competitor".
>
> This also doesn't change the conclusion of our experiments, summarized below:
> - on **bike sharing**, our methods MinSE and AdaMinSE  improve coverage by 2% while having a average length 10% below YK-Base. Recal achieves nominal coverage with set length more than 30% smaller.
> - on **abalone**, differences are more minor: both YK-base and ModelSel undercover by 1%,  Recal is valid and reduces set size by 5% against all competitors, while MinSE and AdaMinSE suffer from conservatism, overcovering by 2% and increasing set size by 10% w.r.t. YK-base
> - on **California housing**, YK-base, ModSel, MinSE, AdaMinSE achieve the same length, but with more than 2% of extra coverage for MinSE and AdaMinSE. Recal has nominal coverage with length 25% smaller than any other competitor.

---

> > ### Comment · Reviewer_6BVL · 2025-08-04
> >
> > Many thanks for your response. As my comments are all addressed I am happy to increase my score.

---

> > ### Comment · Reviewer_6BVL · 2025-08-04
> >
> > Many thanks for your response. As my comments are all addressed I am happy to increase my score.

---

### Official Review · Reviewer_CuX3 · 2025-07-01

**Clarity:** 3
**Significance:** 3
**Originality:** 3
**Rating:** 5
**Confidence:** 3

**Summary:**

This paper presents a compelling framework for selecting among multiple predictive sets based on any criteria, such as selecting the smallest set, while maintaining the coverage guarantees via algorithmic stability. The proposed procedure has coverage guarantees in both batch and online settings. Notably, their method encourages selection on a pointwise basis.

**Questions:**

Could you clarify how AdaMinSE automatically optimises the parameters? I've already looked at the proof, but I might be missing something. It seems to me that the approach is mainly bounding the hyperparameters to eliminate them and using $\tilde{\alpha}$ instead (Update after further reading: It seems that the parameters become variables rather than inputs to the linear program. Still, I'd appreciate your take if you have a clearer or more accurate explanation.)

Additionally, if AdaMinSE is truly optimising the parameters, why does MinSE sometimes perform better? Along the same lines, can you prove that your proposal will outperform any single confidence set?

Do you have any intuition on how to select the prior, especially in the batch setting?

I saw in the appendix that you extended your idea for conditional coverage. Could it be extended for PAC coverage?

**Ethical Concerns:**

["NO or VERY MINOR ethics concerns only"]

**Final Justification:**

The authors have addressed all my concerns, and I still think this is a strong paper. I continue to support acceptance.

**Limitations:**

Yes

**Quality:**

3

**Strengths And Weaknesses:**

The paper is well-written and enjoyable to read. I believe it makes a nice contribution, the core idea and the results are elegant, especially the pointwise selection aspect, which makes this work distinct from other approaches.

However, despite the nice derivations and theoretical results, the proposed approach does not appear to meaningfully outperform the baseline methods, particularly YK-BASE. From what I understand, the method alone does not surpass the baseline unless the recalibration step described in Section 5 is applied. This makes the comparison somewhat unfair, as it relies on an additional calibration set.

However, it seems that the method demonstrates its true strength primarily in the online setting. Could you confirm if this is the case? If it is true, could you give more intuition on this difference?

---

> ### Author Rebuttal · Authors · 2025-07-30
>
> We are very grateful for your review and constructive feedback. We proceed to address your questions below.
>
> **Regarding experimental performance**
>
> The stability-based coverage bound (with the $e^\eta$ factor) can be seen as a worst-case guarantee. As shown in our synthetic examples, this bound is tight in pathological cases but can be conservative in more benign settings. This generality and robustness are precisely what make the framework applicable to the challenging online setting, with no distributional assumptions. However, it can result in overcoverage in less pathological cases.
>
> To address this potential conservativeness in the more structured batch split-conformal setting, we introduced the recalibration procedure (Recal). It circumvents the stability bound by leveraging the rank structure of split-conformal prediction. We want to emphasize that Recal does *not* use an additional calibration set. We split the *single, existing* calibration data budget. Thus, the comparison with baselines is fair in terms of the total data used.
>
> Recal thus indeed systematically outperforms competing methods, achieving valid coverage with up to 30% smaller intervals. However, MinSE and AdaMinSE do also provide some improvement on the validity/efficiency tradeoff. On 2 datasets, they reduce or maintain the set size (by 10% on Bike Sharing) while significantly reducing miscoverage (by more than 2% out of 10%). On the third dataset (Abalone), their average set size is indeed 10% above YK-base, but YK-base simply does not achieve validity, while both MinSE and AdaMinSE achieve coverage 91+%. See detailed recap on experiments at the end.
>
>
> **Regarding batch vs. online settings:** For our online setting, indeed, AdaCOMA can adapt to shifts more quickly than alternative methods that are purely history-based.
>
> **Response to Questions:**
>
> 1.  **How AdaMinSE optimizes parameters:** Your understanding is perfectly correct. Instead of taking $\eta$ and $\tau$ as fixed inputs, AdaMinSE treats them as optimization variables within the linear program. The key intuition is that for any initial miscoverage level $\alpha'$ and a target level $\alpha$, there is a trade-off curve defined by $e^\eta \alpha' + \tau \le \alpha$. For each test point, AdaMinSE automatically finds the point on this curve that minimizes the expected set size for that specific instance, thus adaptively choosing the best tradeoff between $\eta$ and $\tau$.
>
> 2.  **Why MinSE sometimes outperforms AdaMinSE:** This is a subtle but important point. While AdaMinSE is adaptive, its performance depends on the initial miscoverage $\alpha'$ provided. MinSE, in our experiments, uses a fixed $(\eta=1, \tau=0)$. Theoretically, if we were to run AdaMinSE with an initial level $\alpha' = \alpha / e^1$, it would be guaranteed to perform at least as well as this specific MinSE instance. The small performance differences observed in our experiments arise because the $\alpha'$ we chose for AdaMinSE may not have perfectly corresponded to this optimal point for the given dataset, making the fixed-parameter MinSE appear slightly better by chance.
>
> 3.  **Proof of outperforming a single confidence set:** We cannot prove that our method will produce a smaller set than the best single predictor for *every* point. However, our framework is designed to be superior *on average* in heterogeneous settings where no single predictor is uniformly best (as illustrated in our Example 3). In such cases, being able to adaptively switch between predictors pointwise will naturally lead to a smaller average set size compared to being locked into any single predictor.
>
> 4.  **How to select the prior $b$:** We investigated this direction. A principled way to set $b$ is via cross-validation, which leads to a bilevel optimization problem. The outer loop optimizes $b$ to minimize the average selected set size on a validation set, while the inner loop is the MinSE optimization itself. To make the problem easier, we found that adding a negative entropy regularizer to the inner problem makes it strongly convex and much more tractable, admitting efficient solution methods. For simplicity and because the paper was already dense, we used a uniform prior in our experiments. We will add a brief discussion of this approach for setting the prior to the appendix.
>
> 5.  **Extension to PAC coverage:** Thank you for this insightful question. Our current framework provides guarantees on the *expected* miscoverage. PAC-style guarantees, which provide high-probability bounds on the empirical miscoverage rate (e.g., $P(\text{Miscoverage} > \alpha + \epsilon) < \delta$), are indeed a very interesting and powerful extension. This is a non-trivial next step, as it would require analyzing how the randomness from our selection mechanism interacts with the sampling randomness of the test data to affect the concentration of the miscoverage rate. Nonetheless, we believe this extension is achievable.
>
> Thank you again for your feedback, which helps us to improve the paper. We remain available to answer further questions during the discussion period.
>
>
> ---
>
> **Erratum and recap on experiments**
> We wanted to openly disclose that we detected after submission a typo in the code, in file `batch_setting/rescale_residual.py` line 162, `cal_sigma` should be `cal_sigma[m]`. This changes the performance of ModSel, that is ultimately on-par with YK-base (yet never better, in fact YK-base lower bounds the confidence set size of ModSel). Other methods are unaffected. It thus does not change the quality of the "best competitor".
>
> This also doesn't change the conclusion of our experiments, summarized below:
> - on **bike sharing**, our methods MinSE and AdaMinSE  improve coverage by 2% while having a average length 10% below YK-Base. Recal achieves nominal coverage with set length more than 30% smaller.
> - on **abalone**, differences are more minor: both YK-base and ModelSel undercover by 1%,  Recal is valid and reduces set size by 5% against all competitors, while MinSE and AdaMinSE suffer from conservatism, overcovering by 2% and increasing set size by 10% w.r.t. YK-base
> - on **California housing**, YK-base, ModSel, MinSE, AdaMinSE achieve the same length, but with more than 2% of extra coverage for MinSE and AdaMinSE. Recal has nominal coverage with length 25% smaller than any other competitor.

---

> > ### Comment · Reviewer_CuX3 · 2025-08-08
> >
> > Thank you for addressing my concerns.
> >
> > I’m curious to learn more about the optimisation of the prior distribution. Could you please provide additional details? Would incorporating such optimisation affect the theoretical guarantees?
> >
> > Regarding "Why MinSE sometimes outperforms AdaMinSE," then why not run AdaMinSE with the same initial level, which guarantees it will perform at least as well as that specific MinSE instance?

---

> > > ### Author Response · Authors · 2025-08-08
> > >
> > > Thank you for the helpful feedback.
> > >
> > > **On AdaMinSE vs. MinSE Performance**
> > >
> > > You are correct. Theoretically, AdaMinSE with an initial miscoverage level of $\alpha' = \alpha e^{-\eta}$ is guaranteed to perform at least as well as MinSE with parameters $(\eta, \tau=0)$. In our experiments, we tried to show the performance of our method with minimal tuning. Nonetheless, we think this aspect can be further clarified in the paper (at least in text). We are considering adding some experiments to the appendix to compare MinSE against an AdaMinSE instance, where AdaMinSE is using the same stability budget.
> > >
> > > **On Optimizing the Prior Distribution $b$**
> > >
> > > The prior $b$ can be optimized via a bilevel program. The outer loop seeks to find the prior $b^*$ that minimizes the expected selected set size, given by the objective:
> > >
> > > $$E_{X \sim D_{valid}} \left[ \sum_{i=1}^K p_i^*(b, \xi(X)) \lambda(C_i(X)) \right] $$
> > >
> > > The term $p^*(b, \xi(X))$ is the solution to the inner MinSE problem for a given prior $b$, which minimizes the objective $ \sum_{i=1}^{K} p_i \lambda(C_i(X))$ subject to the MinSE constraints.
> > >
> > > This bilevel program may be tackled in a few ways. Modern differentiable frameworks (for example, jaxopt) can backpropagate gradients through (a smoothed variant of) the inner MinSE problem. Doing so allows for solving the objective through first-order methods.
> > >
> > > Another technique is to reformulate the inner LP by relaxing its hard constraints with a differentiable penalty (e.g., using a Bregman divergence or entropy regularizer). This relaxes the problem, making it significantly easier to solve. Alternatively, the optimality conditions of the LP can be used to reformulate the problem.
> > >
> > > Crucially, to preserve the coverage guarantee, this prior optimization must be performed on a validation set held out from the main calibration data, for instance, via a cross-validation split of the training set. However, when the prior tuning is done on such a validation set, all coverage guarantees still hold.
> > >
> > > Again, we sincerely thank you for your engagement in the discussion period.

---

> > > > ### Comment · Reviewer_CuX3 · 2025-08-09
> > > >
> > > > Your clarifications were very helpful, thank you!

---

### Official Review · Reviewer_MLMo · 2025-07-08

**Clarity:** 2
**Significance:** 3
**Originality:** 3
**Rating:** 4
**Confidence:** 4

**Summary:**

This paper takes up an interesting challenge of comparing conformal prediction (CP) sets that are obtained via different learning methods. To this end, the paper proposes a stability-based approach that focuses on coverage for the selected prediction set. It introduces a few stable selection rules, of which MinSE is proved to be optimal. The paper also shows the applicability of this approach to online CP and validates the contributions through a reasonable suite of experiments. It also improves the empirical performance on split CP through a recalibration step.

**Questions:**

Please answer the concerns in the weaknesses above. Below are some additional questions:
* In Corollary 1, how does one find $\hat{S}$ in practice?
* It is interesting to see that when $b$ is a uniform prior (in Sec 3.2), MinSE reduces to deterministically selecting the smallest set. What would this be if $b$ were a prior based on the class frequencies?

Although I have marked the score as BR, I am on the borderline on this paper. The approach is interesting and nicely done; however, the lukewarm results and lack of discussion around the size of prediction sets in later sections of the paper make the final takeaways not as consistent as may be required. I look forward to seeing the authors' responses.

**Ethical Concerns:**

["NO or VERY MINOR ethics concerns only"]

**Final Justification:**

I thank the authors for their responses; they clarified most of my questions. I am increasing my score to a Borderline Accept.

I am also satisfied with the responses from the clarifications in the discussion period, and believe that the paper makes a useful contribution.

(I also want to appreciate the authors for stating and clarifying the code bug issue in a transparent and honest manner.)

**Limitations:**

Yes

**Paper Formatting Concerns:**

* The paper flow seemed to need improvement -- a part of the paper focuses on the size of the prediction set (along with coverage), and a part of the paper focuses only on coverage. The paper did not make this clear enough to parse this distinction.
* Line 130: Should be "...if \hat{S} is $(\eta, \tau, \nu)$-stable..."
* In L163, the paper introduces $\Delta$ without introducing what it stands for. I assume this is the class label simplex; it would be better to make this clear for a reader, however.

**Quality:**

3

**Strengths And Weaknesses:**

Strengths:
+ The idea of using algorithmic stability to get the smallest prediction set in CP that satisfies coverage is novel and interesting.
+ The case studies in Sec 3.3 was nice to see.
+ The additional perspectives on extension to online CP, formulation of adaptive MinSE and recalibration for split CP show a good depth of analysis of the proposed approach.

Weaknesses:
- The paper largely relies on [4] for their work on algorithmic stability and applies those ideas to CP. However, I do think the application to CP is non-trivial and done well.
- For the contribution in Sec 3, a limitation of this work seems to be that you needs to run CP multiple times and get multiple sets/intervals, and then apply this method to find the smallest one. What is the computational overhead of this approach, and what is its practicality? Considering the paper largely focuses on adopting algorithmic stability to CP, it may be important to also comment on the additional complexity introduced.
- Continuing with the above point, the empirical results for this contribution may need to be compared against methods that also have this advantage of seeing multiple prediction sets. Would there be a more fair comparison where the proposed method is compared against other approaches that find the smallest prediction set given multiple CP predictors?
- In Line 139, the paper refers to $\lambda$ as the "size" of set $C_i$, and in the subsequent analysis, $\lambda$ is assumed to be $\in [0,1]$. How would one define such $\lambda$s in practice that actually capture the size of a set (considering the objective of this work is to find the smallest prediction set)? This seems to be key to the stable selection procedure, and I could not find how this was defined in the results section of the main paper too.
- AdaCOMA (Sec 4.1) does not make any comments on the size of the prediction set in this setting. Is it possible to comment on this?
- In the main results (Fig 2), YK-Base seems to do as well/outperform the proposed method on the interval length in Fig 2. While the paper says YK-Base does not guarantee coverage, if $\alpha=0.1$ (as stated in Line 343), YK-Base seems the most calibrated on Bike Sharing and California Housing.
- Also, it is concerning that while the paper compares with references [11] and [12], the datasets used are different from what is used in those papers. This makes it hard to know if the datasets used herein are just favorable for the proposed results, or if these generalize to the datasets used in [11] and [12].
- (Editorial point, but relevant here) It would have been nice to specify $\alpha$ in the caption of Fig 2 itself. It was hard to parse the figure by looking through the text for this value. This issue is exacerbated in the appendix where it was hard to find the confidence level, without which it is hard to interpret coverage results.

---

> ### Author Rebuttal · Authors · 2025-07-31
>
> We thank the reviewer for their thoughtful feedback. Below, we address your valuable comments and questions.
>
> **Response to Weaknesses:**
>
> *  **Regarding Novelty:** We thank you for acknowledging the novelty and non-trivial nature of our contribution.
>
> *  **Regarding Computational Overhead:** We appreciate this practical question. The complexity has two parts: generating $K$ candidate sets, which is somewhat inherent to ensembling-like methods and scales linearly in $K$, and our selection step. The selection, using our algorithms, is efficient. For example, although, we formulated MinSE as a linear program (generally requiring approximately $O(K^3)$ steps to solve), it actually admits a greedy solution with $O(K \log K)$ complexity. This is often negligible compared to the complexity of model inference.
>
> 3.  **Regarding Fairness of Empirical Comparison:** We chose methods from Yang et al. (2024) and Liang et al. (2024) as they are prominent recent works that address the same challenge of selecting from a pool of conformal predictors. As such, the benchmarks are also methods that see and select among a pool of conformal sets.
>
> 4.  **Regarding the Definition of Set Size $\lambda$:** The assumption that $\lambda(C_i^{\alpha}(X)) \in [0,1]$ is made only for notational convenience in Lemma 1 and 2, as noted on Line 157, and is not required for MinSE (and its extension AdaMinSE). As shown by Proposition 1, MinSE algorithm represents the optimal stable algorithm. As such, within the context of this paper, there is generally no motivation to use Lemma 1 and 2 for the selection step and we mainly included them to illustrate different examples of stable selection algorithms. Nonetheless, even for Lemma 1 and 2, the requirement that $\lambda(C_i^{\alpha}(X)) \in [0,1]$ can be relaxed up to a simple rescaling preprocessing step. For all of our experiments in the paper, we used $\lambda$ to be the size the set as measured by the Lebesgue measure. For the experiments in the classification setting, $\lambda$ can be set to the counting measure of different confidence sets.
>
> 5.  **Regarding AdaCOMA and Prediction Set Size:** You are correct that the main text focuses on coverage. The practical advantage in set size is shown in our online experiments in the Appendix, where AdaCOMA produces smaller sets than the COMA baseline by adapting to the current data point $X_t$, while maintaining target coverage.
>
> 6.  **Regarding Comparison with Baselines:** YK-Base indeed performs reasonably well empirically. Nonetheless, it lacks a formal theoretical coverage guarantees. Moreover, on figure 1, Recal systematically provides better results (same coverge with much smaller intervals) and MinSE and AdaMinSE provide shorter or similar intervals with higher coverage on the two of the 3 examples. On the third one YB-base actually undercovers. In addition, in [11], the authors showed an experimental settings where YK-Base undercovers.
>
> 7.  **Regarding Dataset Choice:** This is a fair point. The authors of [11] have open-sourced their code. To facilitate a direct comparison, we commit to add experiments on their synthetic data setting in the final version of our manuscript.
>
> 8.  **Regarding Editorial Point** We will specify the target miscoverage $\alpha=0.1$ in all figure captions.
>
>
>
> **Response to Questions:**
>
> 1.  **How to find $\hat{S}$ in practice?** Corollary 1 is a general result. For implementation, Lemmas 1-3 provide concrete algorithms. Our experiments use the MinSE algorithm from Lemma 3, which is efficient and optimal (Proposition 2). As such, we used MinSE for all of our experiments.
>
> 2.  **What if the prior $b$ in MinSE is non-uniform?** A non-uniform prior $b$ allows incorporating prior knowledge. For a trusted predictor (producing smaller sets) $j$, a higher $b_j$ allows our MinSE algorithm to select it with higher probability. On other hand, a higher $b_j$ does necessitate decreasing another coordinate $b_i$, as $b$ is constrained to the simplex. As such, MinSE will pick a predictor $i$ with less probability (or even never pick it up if $b_i=0$ and $\tau=0$). Thus, a non-uniform prior can be seen as blending prior expert belief with the data-driven results observed during inference time (i.e., after observing the conformal set sizes).  Within this paper, we did not explore how to tune the prior $b$. This is partially due our belief that further tuning of our method may cause unfair comparisons with the benchmark that required no tuning.
>
>
> Finally, we thank you for the paper formatting remarks. We will revise the manuscript to better frame and transition between parts that focus on coverage and parts that focus set size. We are grateful for your feedback and remain eager to answer any follow-up questions or provide further details.
>
>
> ---
>
> **Erratum and recap on experiments**
> We wanted to openly disclose that we detected after submission a typo in the code, in file `batch_setting/rescale_residual.py` line 162, `cal_sigma` should be `cal_sigma[m]`. This changes the performance of ModSel, that is ultimately on-par with YK-base (yet never better, in fact YK-base lower bounds the confidence set size of ModSel). Other methods are unaffected. It thus does not change the quality of the "best competitor".
>
> This also doesn't change the conclusion of our experiments, summarized below:
> - on **bike sharing**, our methods MinSE and AdaMinSE  improve coverage by 2% while having a average length 10% below YK-Base. Recal achieves nominal coverage with set length more than 30% smaller.
> - on **abalone**, differences are more minor: both YK-base and ModelSel undercover by 1%,  Recal is valid and reduces set size by 5% against all competitors, while MinSE and AdaMinSE suffer from conservatism, overcovering by 2% and increasing set size by 10% w.r.t. YK-base
> - on **California housing**, YK-base, ModSel, MinSE, AdaMinSE achieve the same length, but with more than 2% of extra coverage for MinSE and AdaMinSE. Recal has nominal coverage with length 25% smaller than any other competitor.

---

> > ### Comment · Reviewer_MLMo · 2025-08-07
> > **Response to author rebuttal**
> >
> > I thank the authors for their responses; they clarified most of my questions. I am increasing my score to a Borderline Accept, although as mentioned in my original review, I think the paper's analysis can be more thorough. Some specific comments:
> >
> > * I'd recommend having a clear discussion on the computational overhead, and if possible, showing results with the greedy approximation of MinSE. If not, baselines must be given an equivalent compute advantage for a fair comparison.
> > * I'd have appreciated if the results on the synthetic datasets of [11] had been included in the rebuttal period; considering these were synthetic, I assume they may not be time-taking. Once again, my broader pending concern is on fair empirical comparison with baselines here.
> > * I also thank the authors for the open disclosure of the code bug -- could you please share which tables/figures will change due to this? That'd help understand this issue better.

---

> > > ### Author Response · Authors · 2025-08-08
> > >
> > > We sincerely thank the reviewer for their engagement and follow-up questions, which we address below.
> > >
> > > 1. **Regarding computational overhead and the greedy solver for MinSE:** We appreciate the reviewer raising this important practical point. We would like to clarify that the greedy algorithm for our MinSE method is not an approximation; it is an *exact* solver for the corresponding linear program with an efficient $O(K \log K)$ complexity, where $K$ is the number of predictors. We will be sure to clarify this in the final manuscript. The main computational cost for all benchmarked methods comes from training and running inference for the $K$ models. In our experiments (with $K \leq 80$), the additional cost of our selection step was negligible in comparison. Other methods, like ModSel from [11], also involve post-processing steps that carry a similar overhead. In fact, the post-processing steps of ModSel are generally computationally infeasible for regression problems with more than a single-dimensional output.
> > >
> > > 2. **Regarding experiments on datasets from Liang et al. (2024):** During the discussion and rebuttal periods, we prioritized adding new experiments on classification tasks to broaden the paper's scope. While the experiments are synthetic, they are also computationally intensive; based on our preliminary assessments, running them with a large number of predictors (up to 500 in [11] and 80 in our work) would require hundreds of CPU hours. We are fully committed to running these experiments and incorporating the results into the camera-ready version to provide a comprehensive and direct comparison.
> > >
> > > 3. **Regarding the figures impacted:** Thank you for your understanding. To be precise, this bug only affects the performance of the `ModSel` baseline. The results for all our proposed methods (MinSE, AdaMinSE, Recal) and the other baselines (YK-Adjust, YK-Base) are entirely unaffected. The results for the online setting are unaffected. The interval length portion of Figure 2 (and batch setting figures in the appendix) are affected. As noted, after the fix, ModSel's performance becomes similar to that of YK-Base. In fact, by construction, the average set size of YK-Base provides a lower bound for that of ModSel. This correction thus does not alter the paper’s conclusions, as our methods' performance relative to the strongest baselines remains unchanged.
> > >
> > > Again, we sincerely thank the reviewer for their engagement during the discussion period.

---

> > > > ### Comment · Reviewer_MLMo · 2025-08-08
> > > >
> > > > Thank you very much for your clarifications. I am happy with the responses, and stay positive on the paper.

---

### Note · Authors · 2025-08-16

We sincerely thank the reviewers for their constructive feedback and productive engagement during the discussion, which concluded with all reviewers positively assessing our work.

At a high level, we provide stable mechanisms that enable pointwise selection among different conformal sets while preserving validity. Our contributions include designing an optimal selection mechanism, extending the framework to the online setting, and proposing a refined approach tailored to split-conformal prediction. In addition, we present several extensions: an adaptive (tuning-free) variant of the stable mechanism, conditional-coverage guarantees, and tightness and behavioral analyses across multiple illustrative examples.

During the rebuttal period, we clarified multiple points including: the computational overhead of MinSE; the pointwise nature of the selection—performed per test input using that input’s candidate-set sizes—rather than a global or best-on-average aggregation; and Recal’s use of the same calibration budget while mitigating the conservativeness of stability bounds in split-conformal settings. We have already acted on feedback by adding a classification study illustrating pointwise gains and by clarifying various aspects of the proposed methods.

For the camera-ready version, we will include a small set of further results: additional classification experiments; comparison against the mean-ensemble baseline where applicable; and expanded discussion and experiments reflecting the score-aggregation work highlighted by the Area Chair. With the extra page, we will add structural and narrative refinements (clearer flow, captions, conclusion, and limitations) to further improve clarity and quality of the manuscript.

Again, we are grateful for the reviewers’ and the Area Chair’s feedback and guidance.

---

### Decision · Program_Chairs · 2025-09-17

**Decision:**

Accept (poster)

**Comment:**

This paper presents a novel and well-motivated framework for selecting among multiple conformal prediction sets while maintaining coverage guarantees. The reviewers found the proposed stability-based approach to be elegant, technically sound, and a valuable contribution to the field. Initial concerns regarding computational overhead, empirical comparisons to baselines, and certain methodological details were effectively addressed by the authors during the rebuttal period. The authors' clarification of their method's performance and their transparent handling of a minor code bug in a baseline comparison were appreciated. Following the discussion, the reviewers were satisfied with the authors' responses and acknowledged the paper's strengths, viewing it as a solid contribution to the literature.